# Non-cell-autonomous activation of IL-6/STAT3 signaling mediates FGF19-driven hepatocarcinogenesis

Mei Zhou[1], Hong Yang[1], R. Marc Learned[1], Hui Tian[1] & Lei Ling[1]

Hepatocellular carcinoma (HCC), a primary malignancy of the liver, is the second leading cause of cancer mortality worldwide. Fibroblast Growth Factor 19 (*FGF19*) is one of the most frequently amplified genes in HCC patients. Moreover, mice expressing an *FGF19* transgene have been shown to develop HCC. However, the downstream signalling pathways that mediate *FGF19*-dependent tumorigenesis remain to be deciphered. Here we show that FGF19 triggers a previously unsuspected, non-cell-autonomous program to activate STAT3 signalling in hepatocytes through IL-6 produced in the liver microenvironment. We show that the hepatocyte-specific deletion of *Stat3*, genetic ablation of *Il6*, treatment with a neutralizing anti-IL-6 antibody or administration of a small-molecule JAK inhibitor, abolishes FGF19-induced tumorigenesis, while the regulatory functions of FGF19 in bile acid, glucose and energy metabolism remain intact. Collectively, these data reveal a key role for the IL-6/STAT3 axis in potentiating FGF19-driven HCC in mice, a finding which may have translational relevance in HCC pathogenesis.

[1] NGM Biopharmaceuticals, Inc., 333 Oyster Point Boulevard, South San Francisco, California 94080, USA. Correspondence and requests for materials should be addressed to L.L. (email: lling@ngmbio.com).

HCC is the second most common cause of cancer-related deaths worldwide[1]. The pathogenesis of HCC is frequently linked to inflammatory responses triggered by chronic viral infection, alcohol consumption, toxin ingestion and metabolic stress[2–4]. Despite the considerable efforts that have been made in basic and clinical research, the current state of treatment for HCC lags far behind that for many other solid tumours. Consequently, there remains an urgent need to develop effective HCC therapeutics targeting select patient populations.

Over the past decade, multiple studies have delineated a comprehensive landscape of genetic alterations in HCC tumour samples[5]. In particular, focal amplification of chromosome locus 11q13, a region containing fibroblast growth factor 19 (FGF19), was identified as one of the most frequent amplification events in HCC tumours, showing the highest amplitude observed among all genes[5,6]. Notably, amplification of the FGF19 locus is associated with more aggressive tumours, higher risk of recurrence after resection and lower survival rates[7–10]. Studies in mouse models, in which ectopic expression of FGF19 promotes the development of HCC, have further implicated FGF19 as a potential driver of hepatocellular carcinogenesis[11,12], although the mechanism by which FGF19 triggers these events has yet to be resolved.

A multifunctional endocrine hormone, FGF19 plays important roles in regulating bile acid, carbohydrate and energy homoeostasis and liver regeneration[13,14]. Physiological levels of FGF19 act on hepatocytes to limit de novo synthesis of bile acids and protect the liver from these detergent-like molecules[13]. These effects of FGF19 are mediated by a receptor complex comprising fibroblast growth factor receptor 4 (FGFR4) and β-klotho (KLB)[13]. However, the liver-enriched receptor FGFR4 is also thought to mediate the tumorigenic effects of FGF19, since inactivation of FGFR4 via gene knockout or a neutralizing antibody reduces the tumour burden in FGF19 transgenic mice[15]. Conceptually, FGF19 signalling represents a prototypical oncogenic addiction loop, emerging as an attractive molecular target for potential therapeutic intervention in the treatment of HCC.

To that end, a neutralizing anti-FGF19 antibody was developed and shown to effectively block tumorigenesis in HCC models[16]. However, significant safety concerns associated with targeted FGF19 inhibition were revealed during preclinical evaluation of these neutralizing antibodies[17]. As FGF19-FGFR4 signalling plays an important physiological role in the regulation of hepatic bile acid synthesis[13], it is perhaps not surprising that blockade of this pathway with an anti-FGF19 antibody in non-human primates triggered an array of responses consistent with the disruption of bile acid homoeostasis, including severe diarrhoea, liver injury and death[17]. Although FGFR4-specific small molecule inhibitors are in clinical development[18], the potential safety impact of such compounds, especially on bile acid metabolism, has yet to be determined. Therapeutic inhibition of the FGF19-FGFR4 pathway must therefore take into account considerations of both cancer biology and bile acid physiology.

In the present study, we reveal a pivotal role for the signal transducer and activator of transcription 3 (STAT3) and interleukin-6 (IL-6) signalling pathway in FGF19-associated hepatocarcinogenesis. In particular, we demonstrate that blockade of the IL-6/STAT3 axis eliminates FGF19-induced tumour formation without compromising the metabolic functions of FGF19 in regulating bile acid, glucose and energy homoeostasis. In addition, these findings provide evidence that the STAT3/IL-6 pathway represents a potential therapeutic node for treating patients with FGF19-driven HCC.

## Results

**Stat3 is essential for FGF19-induced hepatocarcinogenesis.** Our previous work implicated STAT3 activation in FGF19-mediated signalling pathway[12]. As STAT3 has been previously identified as an oncogenic transcription factor critical for tumour initiation and growth[19–21], we directly examined the role of STAT3 in FGF19-driven tumorigenesis by selectively deleting the Stat3 gene in hepatocytes. Recombinant adeno-associated virus (AAV) carrying Cre recombinase under the control of the hepatocyte-specific, thyroxine-binding globin (TBG) promotor[22] was injected intravenously into mice harbouring LoxP sites flanking exon 18, 19 and 20 of the Stat3 gene (Stat3$^{f/f}$ mice)[23] (Fig. 1a,b). We confirmed the ablation of Stat3 in the resulting hepatocyte-specific Stat3-deficient mice (Stat3$^{\Delta Hep}$) by immuno-histochemical analysis (Fig. 1c).

Next, we examined whether prolonged exposure to FGF19 can induce liver tumours in mice in which the Stat3 gene had been specifically deleted in hepatocytes. Following a single tail vein injection of $1 \times 10^{11}$ genome copies of AAV-FGF19 viral particles, sustained FGF19 expression can be detected for 12 months in the sera of these mice (Supplementary Fig. 1a). As shown in Fig. 1d–f, ~90% of Stat3$^{f/f}$ mice developed hepatocellular carcinomas 12 months after intravenous injection of AAV-FGF19. In contrast, tumour incidence, tumour multiplicity and tumour size were all markedly reduced in age-matched Stat3$^{\Delta Hep}$ mice, even after 12 months of continuous exposure to FGF19. Histological analysis of mice ectopically expressing FGF19 revealed that glutamine synthetase-positive HCC tumours developed exclusively in Stat3$^{f/f}$ mice, but were not observed in Stat3$^{\Delta Hep}$ mice (Fig. 1d,g, and Supplementary Fig. 1b), suggesting that Stat3 deficiency in hepatocytes prevents HCC initiation and progression in response to FGF19. Furthermore, the FGF19-induced tumours observed to develop in Stat3$^{f/f}$ mice were highly proliferative, as demonstrated by immunohistochemical staining with antibodies against Ki-67 (Supplementary Fig. 1c). FGF19 expression increased liver-to-body weight ratios in Stat3$^{f/f}$ mice, but not in Stat3$^{\Delta Hep}$ mice (Fig. 1h). Concentrations of FGF19 in the sera of these mice were measured by ELISA at the end of the study, and found to be $292 \pm 29$ ng ml$^{-1}$ and $275 \pm 47$ ng ml$^{-1}$ in Stat3$^{f/f}$ and Stat3$^{\Delta Hep}$ genotypes, respectively (Fig. 1i).

STAT3 confers its protumorigenic property, at least in part, by inducing the transcription of genes important for suppression of apoptosis, such as Survivin (Birc5), Bcl-x$_L$ (Bcl2l1), and cell proliferation, such as Cyclins[20,21]. Marked elevation of Birc5, Bcl2l1 and Ccnd1 messenger RNA (mRNA) levels was observed in livers from FGF19-expressing Stat3$^{f/f}$ mice (Supplementary Fig. 1d). In contrast, mRNA levels of these anti-apoptotic genes were unchanged by ectopic FGF19 expression in livers of Stat3$^{\Delta Hep}$ mice. In addition, markers of cell cycle progression (Ccna2, Ccnb1, Ccnb2), cell proliferation (Mki67, Pcna) and HCC (α-fetoprotein or Afp)[24], were profoundly induced by FGF19 expression in livers from Stat3$^{f/f}$ mice, but not Stat3$^{\Delta Hep}$ mice (Supplementary Fig. 1e and f).

Collectively, we conclude from these observations that hepatocellular ablation of Stat3 blocks the initiation and progression of FGF19-dependent HCC formation.

**Stat3 is dispensable for FGF19-regulated metabolism.** As part of the gut-liver axis that controls bile acid metabolism, FGF19 suppresses de novo bile acid synthesis by inhibiting expression of key enzymes in the bile acid synthetic pathway[13]. We analysed effects of ectopic FGF19 expression on bile acid synthetic enzymes in the presence or absence of hepatocellular Stat3 (Fig. 2a). FGF19 expression in both Stat3$^{f/f}$ and Stat3$^{\Delta Hep}$ mice

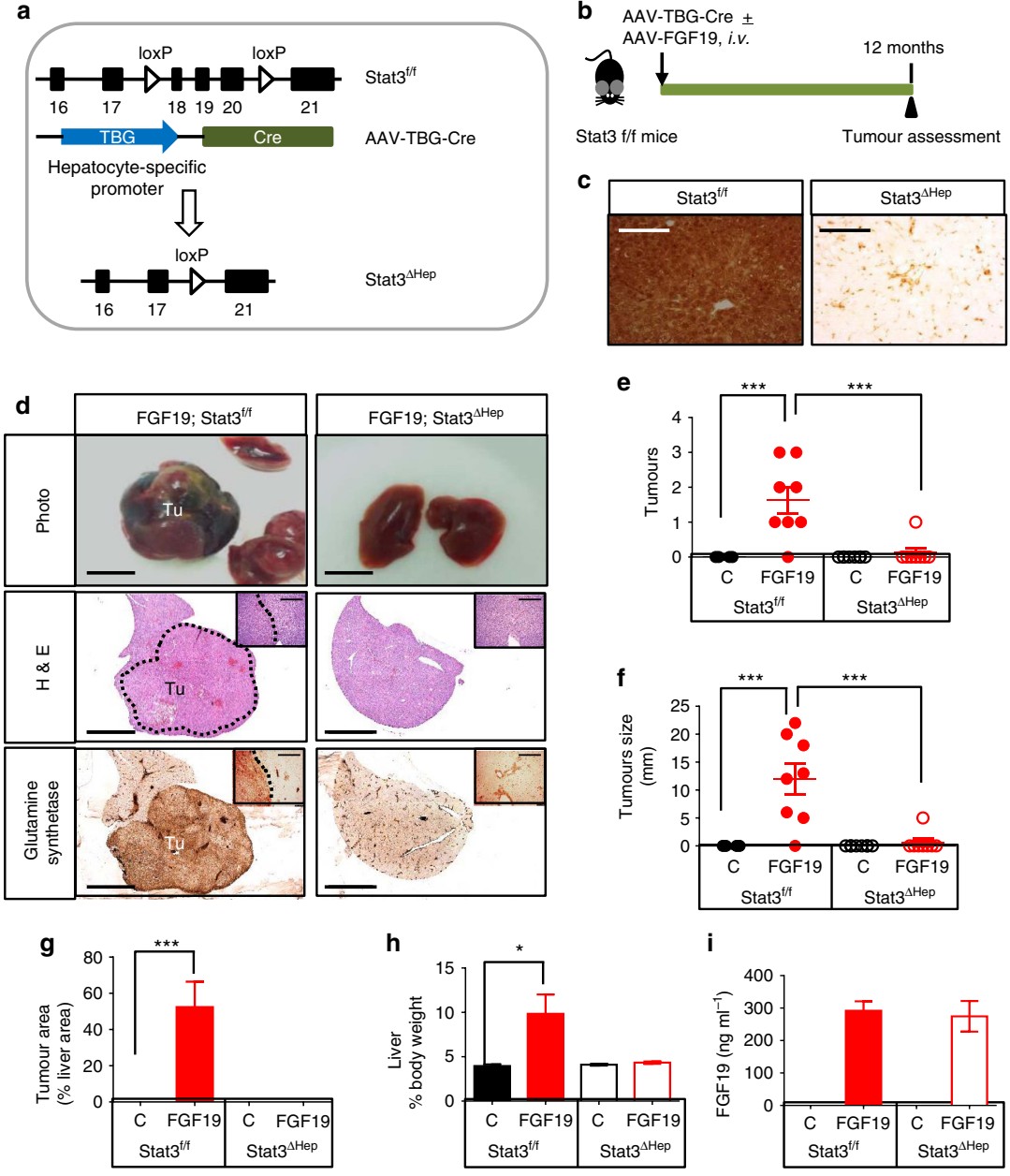

**Figure 1 | Hepatocyte-specific ablation of STAT3 eliminates FGF19-associated tumorigencity.** (**a**) Schematic representation of generating Stat3^ΔHep mice with AAV-mediated delivery of Cre recombinase. The thyroxine binding globin (TBG) promoter drives hepatocyte-specific Cre expression. Exons in Stat3 gene are labelled. (**b**) Study design. 14 to 18 week-old Stat3^f/f mice received a single tail vein injection of AAV-FGF19 (n = 8), or a combination of AAV-FGF19 and AAV-TBG-Cre (n = 8), or a control virus GFP (C) (n = 5). Mice were killed 12 months after AAV administration for liver tumor analysis. (**c**) Confirmation of hepatocellular STAT3 ablation by immunohistochemical staining with anti-STAT3 followed by DAB substrates (brown colour). Abundant STAT3 proteins were detected in livers from Stat3^f/f mice, but not Stat3^ΔHep mice. Note residual STAT3 expression in non-parenchymal cell in Stat3^ΔHep mice. Scale bars, 100 μm. (**d**) FGF19 induces HCC in Stat3^f/f mice (n = 8), but not Stat3^ΔHep mice (n = 8). Shown are representative macroscopic view, and liver sections stained with H&E or anti-glutamine synthetase. DAB substrates (brown colour) were used for immunohistochemistry. Tu, tumours. Scale bars, 5 mm. Higher magnifications of liver sections are shown in insets. (**e**) Liver tumour multiplicity. Dots in scatterplot represent individual animals. (**f**) Liver tumour size recorded as maximum tumour diameter in each mouse. (**g**) Quantification of glutamine synthetase-positive tumour area as a percentage of total liver area. (**h**) Liver weights from mice of the indicated genotypes. (**i**) Circulating levels of FGF19 at the end of the study. Values are mean ± s.e.m. ***$P < 0.001$, *$P < 0.05$ by unpaired, two-tailed t-test of indicated groups.

resulted in similar reductions in the mRNA levels of Cyp7a1, encoding the first and rate-limiting enzyme responsible for the classic bile acid synthetic pathway (77% and 87% reduction, respectively; Fig. 2b), and Cyp8b1, which encodes an enzyme involved in cholic acid synthesis (95% and 88% reduction, respectively, in Stat3^f/f and Stat3^ΔHep mice; Fig. 2c). Thus,

hepatocellular Stat3 appears to be dispensable for FGF19-mediated regulation of bile acid metabolism.

The biological actions of FGF19 have been reported to extend beyond the regulation of bile acid homoeostasis. For example, FGF19 has been shown in rodent models of metabolic diseases to suppress hepatic gluconeogenesis by

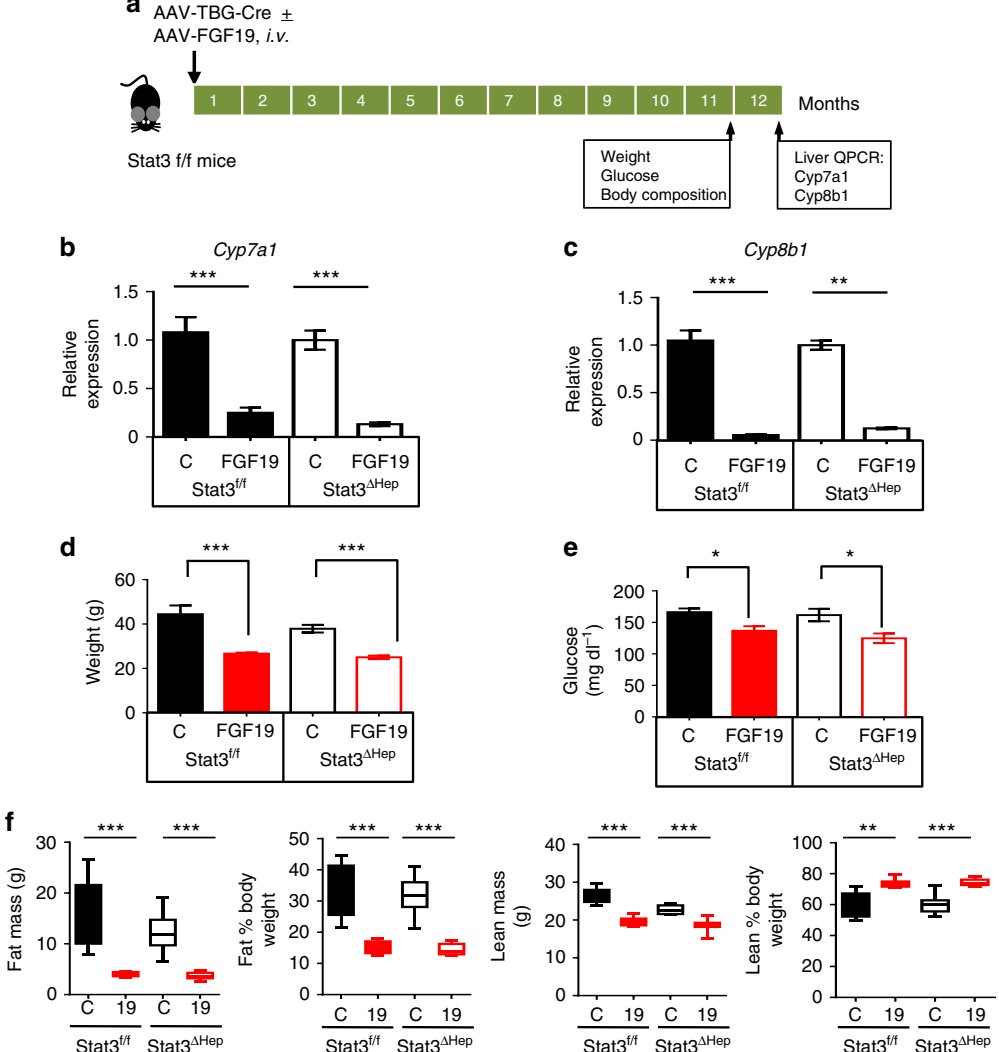

**Figure 2 | Loss of STAT3 in hepatocytes does not impair FGF19-dependent metabolic improvements.** (**a**) Study design. 14 to 18 week-old $Stat3^{f/f}$ mice received a single tail vein injection of AAV-*FGF19* ($n = 8$), or a combination of AAV-*FGF19* and AAV-TBG-Cre ($n = 8$), or a control (C) virus ($n = 5$). Mice were killed 12 months after AAV administration to determine expression of *Cyp7a1* and *Cyp8b1* in the liver. Body weight, blood glucose, and body composition were measured 1 month before euthanasia. (**b**) Hepatic *Cyp7a1* mRNA levels. Data are normalized to housekeeping gene GAPDH, and are relative to the expression in $Stat3^{f/f}$ control mice. (**c**) Hepatic *Cyp8b1* mRNA levels. (**d**–**f**) Body weight (**d**), plasma glucose (**e**) and body composition (**f**) were determined in live animals. Values are mean ± s.e.m. ***$P < 0.001$, **$P < 0.01$, *$P < 0.05$ by unpaired, two-tailed *t*-test of indicated groups.

regulating the peroxisome proliferator-activated receptor gamma coactivator 1-α (PGC-1α)/cyclic AMP (cAMP) response element binding protein (CREB) pathway[25], increase fatty acid oxidation and energy expenditure[26] and improve glucose effectiveness through the hypothalamus-pituitary-adrenal axis[27,28]. Consistent with these reports, the weight- and glucose-lowering effects of FGF19 were fully evident in $Stat3^{f/f}$ mice (Fig. 2d,e). Importantly, similar results were obtained in $Stat3^{\Delta Hep}$ mice, implying that FGF19 functions in a $Stat3$-independent manner to regulate both body weight and blood glucose. Furthermore, hepatocyte-specific $Stat3$ deficiency had no effect on FGF19-mediated changes in body composition, as fat and lean mass were indistinguishable in $Stat3^{f/f}$ and $Stat3^{\Delta Hep}$ mice expressing FGF19 (Fig. 2f). Similarly, liver steatosis was improved by FGF19 treatment to a similar degree in $Stat3^{f/f}$ and $Stat3^{\Delta Hep}$ mice (Supplementary Fig. 2).

Therefore, while critical for FGF19-dependent HCC formation, hepatocellular $Stat3$ is not required for FGF19 to regulate bile acid biosynthesis or restore metabolic homoeostasis.

**Non-cell autonomous activation of hepatic STAT3 by FGF19.** We further examined STAT3 activation in response to FGF19 treatment in mouse livers *in vivo*, and in isolated hepatocytes *in vitro*. Located in the cytoplasm of cells, STAT3 is a latent transcription factor. On activation by upstream kinases, Y705, a conserved tyrosine residue, is phosphorylated, leading to dimerization *via* the Src Homology 2 (SH2) domain, nuclear translocation, DNA binding and activation of target gene expression in the cell nucleus[20,21]. Diabetic *db/db* mice (on BKS background) were used for these studies, as this mouse strain showed comparatively high rates of tumour penetrance and short tumour latency, when exposed to FGF19 (ref. 12).

As shown by immunoblotting of liver lysates prepared from *db/db* mice, STAT3 was phosphorylated at tyrosine residue 705 (pSTAT3$^{Y705}$) following administration of recombinant FGF19 protein (Fig. 3a,b). No phosphorylation of this conserved tyrosine residue was detected in other STAT isoforms (Supplementary Fig. 3a). Consistent with previous reports[29], FGF19 also induced phosphorylation at threonine-202 and

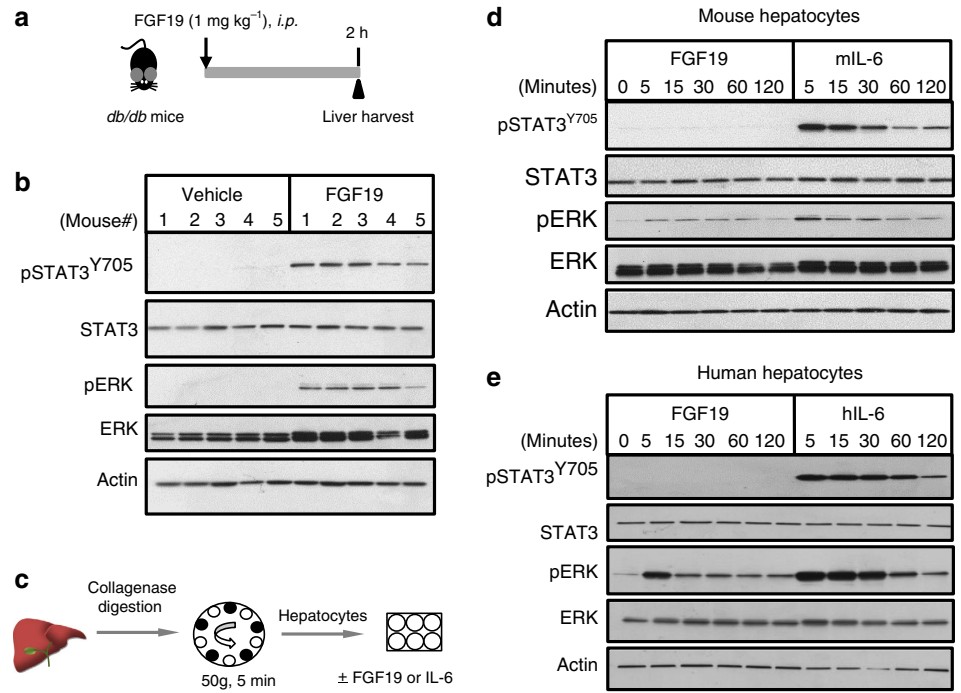

**Figure 3 | Non-cell-autonomous activation of hepatocellular STAT3 by FGF19.** (**a**) *in vivo* study design. 11–12-week old *db/db* mice received a single intraperitoneal injection of 1 mg kg$^{-1}$ FGF19 or vehicle (saline), and livers were harvested 2 h post dose ($n = 5$ per group). (**b**) Immunoblot analysis of pSTAT3$^{Y705}$ in liver lysates of *db/db* mice treated with recombinant FGF19 protein. Anti-total-STAT3 and anti-β-actin serve as loading control. pERK and total ERK levels were also determined. (**c**) Primary hepatocytes were isolated by collagenase digestion followed by low-speed centrifugation and plating onto collagen-coated plates. (**d**) Lack of pSTAT3$^{Y705}$ activation in primary mouse hepatocytes by FGF19. Cell lysates were prepared at the indicated time points following FGF19 stimulation and analysed for phosphorylation of the various proteins. Mouse IL-6 (mIL-6) was included as a positive control. (**e**) Lack of pSTAT3$^{Y705}$ activation by FGF19 in primary human hepatocytes. Human IL-6 (hIL-6) was included as a positive control.

tyrosine-204 of extracellular signal-regulated kinases ERK1 and ERK2 in the liver (Fig. 3b).

To investigate the possibility that FGF19 acts directly on hepatocytes to activate STAT3 signalling, immunoblot analysis of cell lysates was carried out following FGF19 stimulation of primary cultures of hepatocytes. Hepatocytes were purified from livers and confirmed to express FGF19 receptors FGFR4 and KLB (Fig. 3c and Supplementary Fig. 3b). In marked contrast to the observations from the *in vivo* studies, treatment of isolated mouse hepatocytes with recombinant FGF19 protein failed to promote STAT3 phosphorylation (Fig. 3d). Similarly, the levels of pSTAT3$^{Y705}$ were unchanged from baseline in response to FGF19 treatment of purified primary human hepatocytes (Fig. 3e). As a positive control, robust elevated phosphorylation of pSTAT3$^{Y705}$ was detected following IL-6 treatment of primary hepatocytes. These data suggest that the mechanism by which FGF19 activates STAT3 in hepatocytes may not be direct, and furthermore, that factors other than FGF19 may also be required.

In contrast, phosphorylation of ERK1 and ERK2 was detected in primary cultures of both mouse and human hepatocytes treated with FGF19 (Fig. 3d,e), similar to observations *in vivo*. Taken together, these data are consistent with a cell-autonomous mechanism of ERK activation by FGF19.

These paradoxical *in vivo* and *in vitro* observations support a non-cell-autonomous activation of STAT3 by FGF19, and suggest the possibility that FGF19-triggered STAT3 phosphorylation is mediated by components residing in the liver microenvironment.

**Non-cell autonomous induction of proliferation by FGF19.** To determine whether FGF19 directly regulates hepatocyte

proliferation in a cell-autonomous manner, we compared the proliferative effects of FGF19 *in vivo* and in isolated hepatocytes by assessing the incorporation of 5-bromo-2-deoxyuridine (BrdU) into newly synthesized DNA.

When administered to mice for six days through an osmotic pump (Fig. 4a), recombinant FGF19 protein induced incorporation of BrdU into hepatocytes as evidenced by both immunohisto-chemical staining (Fig. 4b), and flow cytometry analysis (Fig. 4c,d). At a dose of 20 μg per mouse per day, livers from FGF19-treated mice contain significantly more BrdU-positive cells (9.2 ± 1.5%) than livers harvested from mice treated with vehicle alone (1.7 ± 0.2%) (Fig. 4d). Consistent with this result, levels of proliferating cell nuclear antigen (PCNA) and Ki-67 expression were higher in FGF19-treated mice (Supplementary Fig. 4a). In agreement with a previous report[11], FGF19 induced BrdU incorporation in peri-central hepatocytes marked by glutamine synthetase expression (Supplementary Fig. 4b). As determined by ELISA, plasma concentrations of FGF19 were 5.2 ± 1.2 ng ml$^{-1}$ at the end of the study (Fig. 4e).

The proliferative effects of FGF19 observed *in vivo* contrast sharply with *in vitro* observation. No increase in BrdU signal was detected when primary cultures of hepatocytes isolated from mouse livers were incubated with increasing concentrations of FGF19 (Fig. 4f). Similarly, no increase in cell proliferation was observed when purified human hepatocytes were treated with recombinant FGF19 (Fig. 4g). In contrast, BrdU incorporation was significantly increased when cultured hepatocytes were stimulation with hepatocyte growth factor (HGF). The apparent disconnect between the *in vitro* and *in vivo* effects of FGF19 on hepatocyte proliferation parallels findings regarding STAT3 activation, as described in Fig. 3.

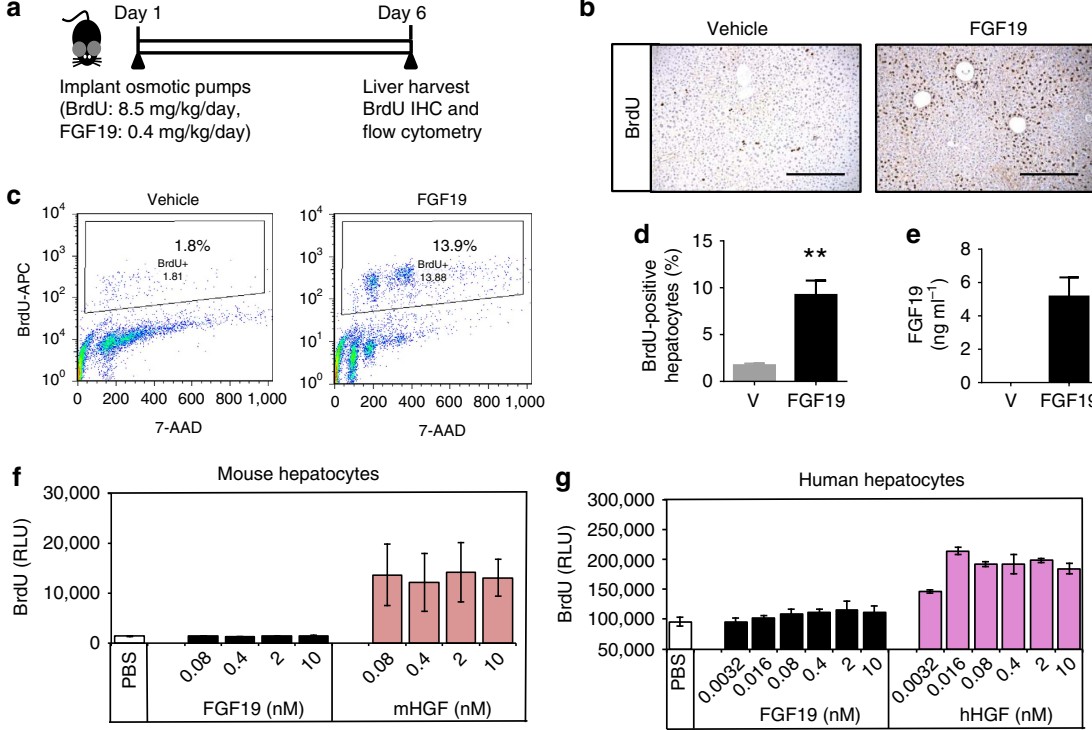

**Figure 4 | Non-cell-autonomous promotion of hepatocellular proliferation by FGF19.** (**a**) *in vivo* study design. Mice ($n = 5$ per group) were implanted with osmotic pumps releasing BrdU (8.5 mg kg$^{-1}$ day$^{-1}$) and FGF19 protein (0.4 mg kg$^{-1}$ day$^{-1}$). Livers were harvested 6 days post implant. (**b**) BrdU incorporation in liver *in vivo* was revealed by staining with anti-BrdU antibody followed by DAB substrates (brown colour). Sections were counterstained with hematoxylin. Scale bars, 100 μm. (**c**) Flow cytometry analysis of BrdU-labelled livers. Representative histograms from hepatocytes stained with anti-BrdU-APC and 7-AAD are shown. (**d**) BrdU-positive hepatocytes as a percentage of total hepatocytes from vehicle (V)-treated or FGF19-treated mice were quantified by flow cytometry. (**e**) Circulating FGF19 levels on day 6 post-implant of osmotic pumps. (**f**) Lack of proliferative effects of FGF19 on primary mouse hepatocytes. Primary cultures of hepatocytes isolated from mouse liver were incubated with recombinant FGF19 protein at indicated concentrations for 48 h, and BrdU was added during the last 24 h of incubation. BrdU incorporation was determined using a luminescence method. Mouse hepatocyte growth factor (mHGF) was included as a positive control. RLU, relative luminescence unit. (**g**) Lack of proliferative effects of FGF19 on primary human hepatocytes. Primary cultures of hepatocytes isolated from human liver were incubated with recombinant FGF19 protein. Human hepatocyte growth factor (hHGF) was included as a positive control. Data are represented as mean ± s.e.m. **$P < 0.001$ versus control group by unpaired, two-tailed *t*-test.

Taken together, these observations suggest that FGF19-induced hepatocyte proliferation *in vivo* might involve a non-cell autonomous mechanism, suggesting a potential interplay between hepatocytes and the non-parenchymal microenvironment.

**Secreted factor(s) mediating activation of STAT3 by FGF19.** These results prompted us to identify the secreted factor(s) mediating FGF19-induced STAT3 activation *in vivo*. Cytokines that activate the gp130 receptor, such as IL-6, IL-11, leukemia inhibitory factor (LIF), oncostatin M (OSM), ciliary neurotrophic factor (CNTF) and cardiotrophin-1 (CTF1), are well-known stimulators of STAT3 signalling[30]. As a first step, we measured hepatic expression of these gp130 ligands in FGF19-treated *db/db* mice. Notably, hepatic mRNA levels of IL-6 increased significantly following administration of FGF19 in *db/db* mice, correlating with STAT3 activation in these mice (Fig. 5a). In contrast, the mRNA levels of IL-11, LIF, OSM, CNTF and CTF1 in mice treated with FGF19 remained unchanged or were undetectable, suggesting that the involvement of these cytokines in this context was less likely (Fig. 5b). In addition, FGF19 administration failed to induce hepatic expression of a panel of growth factors and cytokines associated with increased pSTAT3 signalling, including epidermal growth factor (EGF)[31], IL-10 (ref. 32), IL-21 (ref. 33), IL-22 (ref. 34) and IL-31 (ref. 35) (Fig. 5c). As expected, levels of *Cyp7a1*

mRNA were efficiently suppressed by acute FGF19 treatment in the livers of these mice (Fig. 5d).

To examine whether IL-6 mediates FGF19-induced STAT3 activation in the liver, we injected *db/db* mice intraperitoneally with a neutralizing antibody against mouse IL-6 before FGF19 administration. As demonstrated by immunoblot analysis of total liver lysates, marked reduction of FGF19-triggered STAT3 phosphorylation was observed by pretreatment with anti-IL-6 antibodies (Fig. 5e). No effects on STAT3 activation were observed, however, when mice were pretreated with an isotype control antibody. These results indicate that the non-cell-autonomous activation of STAT3 by FGF19 in mice is mediated by a secreted factor, IL-6.

Next, we investigated the cellular source of IL-6. We isolated non-parenchymal cells from mouse livers to measure IL-6 expression by intracellular cytokine staining (Supplementary Fig. 5a and Fig. 5f). Flow cytometry analysis revealed strong IL-6 expression in CD45-positive liver infiltrating immune cells, whereas IL-6 was undetectable in CD45-negative cells after FGF19 stimulation. Specifically, myeloid cells including Kupffer cells (CD45$^+$CD11b$^+$F4/80$^+$), neutrophils (CD45$^+$CD11b$^+$Ly6G$^+$) and NK cells (CD45$^+$NK1.1$^+$) were the major producers of IL-6, but little or no IL-6 could be detected in T (CD45$^+$CD3$^+$) or B cells (CD45$^+$CD19$^+$) (Fig. 5g and Supplementary Fig. 5b and c).

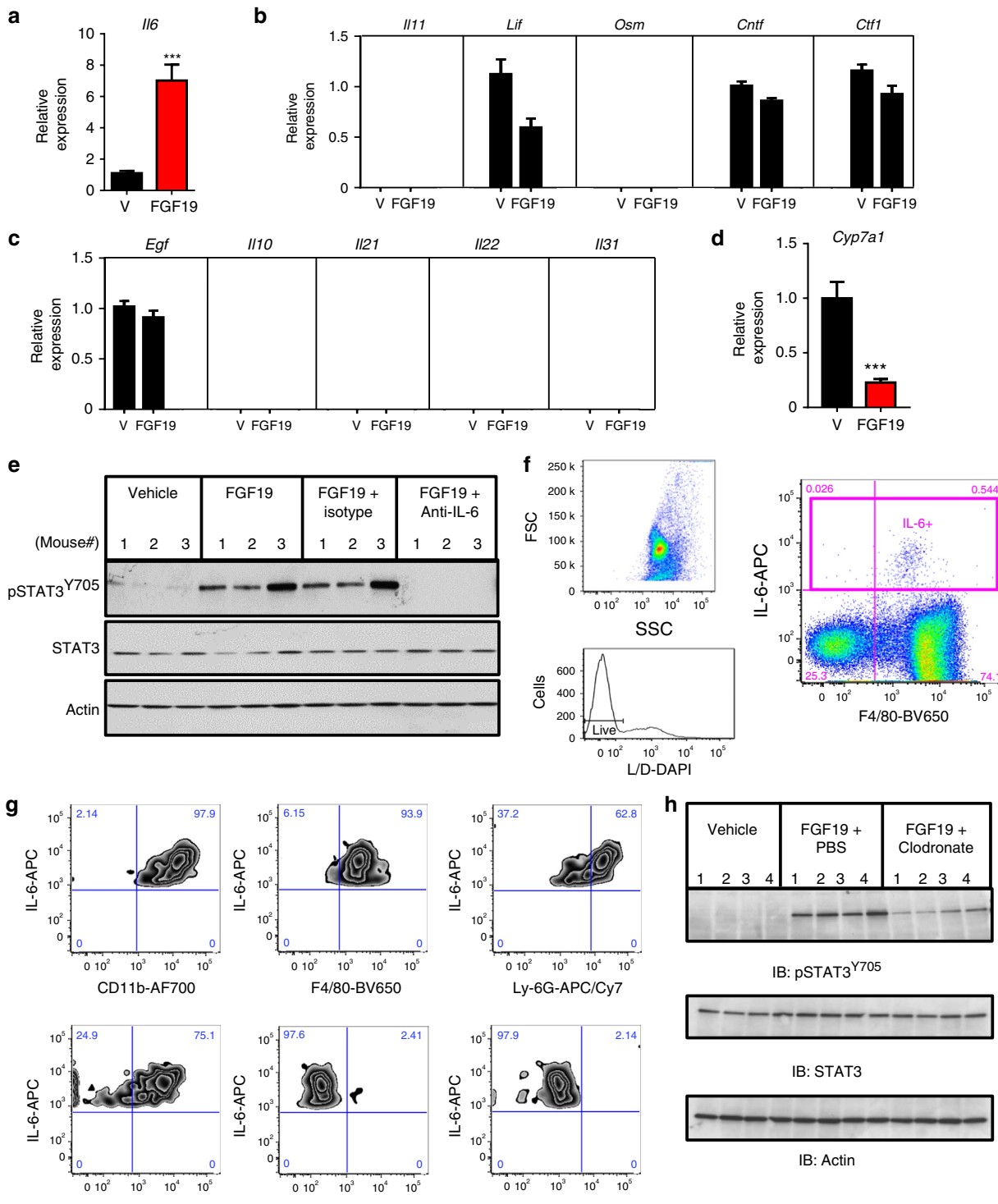

**Figure 5 | Identification of secreted factor(s) mediating non-cell-autonomous activation of STAT3 by FGF19.** (**a**) Hepatic IL-6 mRNA is induced in FGF19-treated *db/db* mice. 11~12-week old *db/db* mice received an intraperitoneal injection of 1 mg kg$^{-1}$ FGF19 or vehicle (V), and livers were harvested 2 h post dose ($n = 5$ per group). (**b**) Lack of induction of canonical pSTAT3-activating cytokines, such as IL-11, LIF, OSM, CNTF and CTF-1, in FGF19-treated mice. (**c**) Lack of induction of additional pSTAT3-activating growth factors and cytokines, such as EGF, IL-10, IL-21, IL-22 and IL-31, in FGF19-treated mice. (**d**) Suppression of hepatic Cyp7a1 mRNA in FGF19-treated mice. (**e**) Blocking antibody against mouse IL-6 abolishes pSTAT3$^{Y705}$ activation by FGF19. 11–12-week old *db/db* mice were injected intraperitoneally with a neutralizing anti-IL-6 antibody before FGF19 administration, and livers were harvested 2 h post FGF19 dose ($n = 3$ per group). (**f**) Intracellular IL-6 cytokine staining of non-parenchymal cells analysed by flow cytometry. Representative forward scatter (FSC) and side scatter (SSC) plot of ungated cells is shown. Live cells were identified based on Live/Dead-DAPI staining (L/D). IL-6-positive cell gate (IL-6 +) was used for subsequent analysis. (**g**) Co-localization of IL-6 with markers of myeloid cells (CD11b +), Kupffer cells (F4/80 +), neutrophils (Ly6G +), NK cells (NK1.1 +), but not T cells (CD3 +) or B cells (CD19 +). (**h**) Depletion of Kupffer cells reduces pSTAT3$^{Y705}$ activation by FGF19. 11–12-week old *db/db* mice ($n = 4$ per group) were injected intravenously with clodronate liposomes (to deplete Kupffer cells) or PBS liposomes. Livers were harvested 2 h post 1 mg kg$^{-1}$ FGF19 dose. Values are mean ± s.e.m. ***$P < 0.001$ by unpaired, two-tailed *t*-test.

To investigate further which immune cell type is responsible to FGF19-driven STAT3 activation, we injected *db/db* mice intravenously with clodronate liposomes, which deplete Kupffer cells from the liver. As shown in Fig. 5h, clodronate liposome treatment inhibited FGF19-induced STAT3 phosphorylation, whereas injection of PBS liposomes had no effect. In contrast, depletion of neutrophils or CD8[+] T cells had minimal effects on FGF19-induced STAT3 phosphorylation (Supplementary Fig. 5d and e).

Collectively, these observations suggest that innate immune cells, and Kupffer cells in particular, are the primary source of IL-6 in response to FGF19-dependent STAT3 activation in hepatocytes, and serve as experimental evidence to define a non-cell-autonomous mechanism for FGF19-triggered hepatocellular oncogenesis.

**Deletion of IL-6 prevents FGF19-induced HCC development.** To further determine the functional significance of IL-6 in FGF19-induced HCC progression, we injected AAV-*FGF19* intravenously into wild type (*Il6*[+/+]) or *Il6*-deficient (*Il6*[−/−]) mice, and evaluated HCC development 12 months post AAV administration (Supplementary Fig. 6a). As expected, prolonged exposure to FGF19 induced HCC formation, as evidenced by large, glutamine synthetase-positive liver tumours in *Il6*[+/+] mice (Fig. 6a). In contrast, genetic ablation of *Il6* resulted in a significant decrease in FGF19-induced tumour multiplicity (Fig. 6b), tumour size (Fig. 6c) and glutamine synthetase-positive tumour area (Fig. 6d, and Supplementary Fig. 6b). Moreover, FGF19 expression increased liver-to-body weight ratios in *Il6*[+/+] mice, but not in *Il6*[−/−] mice (Fig. 6e). These observations regarding the effects of FGF19 in *Il6*-deficient mice are remarkably similar to those described in mice with hepatocellular deficiency of STAT3 (Fig. 1).

As determined by ELISA, circulating levels of FGF19 in *Il6*[−/−] mice were detected at levels equivalent to or greater than in *Il6*[+/+] mice, thereby excluding reduced FGF19 levels as a cause of the diminished tumorigenesis (Fig. 6f). *Il6* deficiency also reduced hepatic mRNA levels of *Stat3* target genes (*Birc5, Bcl2l1* and *Ccnd1;* Supplementary Fig. 6c), cyclins (*Ccna2, Ccnb1, Ccnb2;* Supplementary Fig. 6d), and markers of proliferation (*Mki67;* Supplementary Fig. 6e) and HCC (*Afp;* Supplementary Fig. 6e) in livers from FGF19-expressing mice, further demonstrating the role of IL-6 in mediating FGF19-driven tumour growth.

IL-6 signalling in liver parenchymal cells has been reported to suppress hepatic inflammation and improve systemic insulin action through receptor IL-6Rα specifically expressed on hepatocytes[36]. To determine whether genetic ablation of *Il6* has any effect on FGF19-mediated metabolic action, we monitored body weight, plasma glucose and insulin levels in these mice (Supplementary Fig. 6a). No significant differences in these parameters were detected in FGF19-treated *Il6*[+/+] and *Il6*[−/−] mice (Fig. 6g, Supplementary Fig. 6f and g). Lack of IL-6 expression had no effect on FGF19-triggered repression of either *Cyp7a1* or *Cyp8b1* (Fig. 6h,i). In addition, FGF19 efficiently reduced body fat composition in both *Il6*[+/+] and *Il6*[−/−] mice (Supplementary Fig. 6h). Therefore, IL-6 in mice appears to be dispensable for FGF19-controlled effects on energy homoeostasis or glucose and bile acid metabolism.

In summary, these results indicate that IL-6 produced in the liver microenvironment is important for tumour development initiated by FGF19.

**Inhibition of STAT3/IL-6 axis abolishes FGF19-driven HCC.** We next tested the *in vivo* efficacy of inhibitors of the STAT3/IL-6 pathway in engineered mouse models of FGF19-driven HCC. We have previously reported that prolonged exposure to FGF19

delivered *via* recombinant AAV induces HCC in *db/db* and multidrug resistance 2 (MDR2)-deficient mice (*Mdr2*[−/−])[12,37]. Whereas the diabetic *db/db* mouse model allows us to simultaneously study anti-diabetic and tumorigenic effects of FGF19, the *Mdr2*[−/−] mice, an animal model of chronic liver disease, enables simultaneous interrogation of the hepatoprotective, as well as, HCC-promoting actions of FGF19. These models may be clinically relevant in the context of tumorigenesis, as higher HCC incidence was reported with concomitant diabetes or chronic liver disease[38].

Suppressor of cytokine signalling 3 (SOCS3) binds to the gp130 receptor and regulates IL-6/STAT3 signalling through a negative feedback loop[39]. In the following study, we evaluated whether SOCS3 can serve as an endogenous inhibitor of STAT3 to prevent FGF19-induced HCC. To this end, *db/db* mice were co-administered with AAV-*FGF19* and AAV-SOCS3 at a 1:10 ratio intravenously (Fig. 7a). On the basis of both macroscopic and histological observations, *db/db* mice were shown to develop HCC by 24 weeks after AAV-*FGF19* injection. Co-expression of SOCS3 significantly reduced FGF19-associated liver tumour formation (Fig. 7b, Supplementary Fig. 7a and b), whereas SOCS3 had no impact on FGF19-mediated effects on haemoglobin A1c (HbA1c)- (Fig. 7c) and bile acid-lowering (Supplementary Fig. 7c). Similar circulating FGF19 levels were detected in both groups of mice (Supplementary Fig. 7d).

Activation of STAT3 depends on the phosphorylation of tyrosine residue 705 by upstream kinases, such as Janus kinases[40]. Tofacitinib is a small molecule pan-JAK inhibitor, and an FDA-approved treatment for rheumatoid arthritis[41]. To test whether tofacitinib can inhibit FGF19-dependent tumour formation, we injected *db/db* mice with AAV-*FGF19*, and 4 weeks later, began administration of tofacitinib in the diet (Fig. 7d). Notably, treatment with tofacitinib significantly reduced the number of macroscopically detectable tumours and average tumour load in FGF19-expressing mice (Fig. 7e, Supplementary Fig. 7e and f). However, tofacitinib administration had no impact on the ability of FGF19 to normalize blood levels of glucose (Fig. 7f), HbA1c (Supplementary Fig. 7g) and bile acids (Supplementary Fig. 7h). Circulating FGF19 levels were similar in both groups of mice (Supplementary Fig. 7i).

Finally, we examined whether targeted inhibition of IL-6 can slow FGF19-driven HCC development (Fig. 7g). Since IL-6 is an important mediator of inflammatory diseases, neutralizing antibodies targeting IL-6 or the IL-6 receptor, including siltuximab and tocilizumab, have been developed and approved to treat Castleman's disease and rheumatoid arthritis[42,43]. For the purpose of our study, anti-IL-6 antibody treatment (10 mg kg[−1], intraperitoneally once every week, for a total of 10 weekly treatments) was initiated after 14 weeks of continuous FGF19 exposure in *Mdr2*[−/−] mice (Fig. 7g). Although *Mdr2*[−/−] mice injected with AAV-*FGF19* exhibited significant improvements in serum liver enzyme levels, with reductions observed for alkaline phosphatase (ALP), alanine aminotransferase (ALT) and aspartate aminotransferase (AST) (Supplementary Fig. 7j), the prolonged FGF19 exposure induced liver tumour formation in these mice (Fig. 7h,i). We observed completely healthy livers free of tumours when FGF19-expressing *Mdr2*[−/−] mice were treated with anti-IL-6, whereas those treated with isotype control antibodies develop HCC (Fig. 7h,i and Supplementary Fig. 7k). Importantly, treatment of *Mdr2*[−/−] mice with anti-IL-6 antibodies abolished hepatocarcinogenesis, without altering the anti-cholestatic effects mediated by FGF19, including reducing the serum levels of ALP (Fig. 7j), ALT (Supplementary Fig. 7l), AST (Supplementary Fig. 7m) and bile acids (Fig. 7k). Similar circulating FGF19 levels were observed in all groups of mice (Supplementary Fig. 7n).

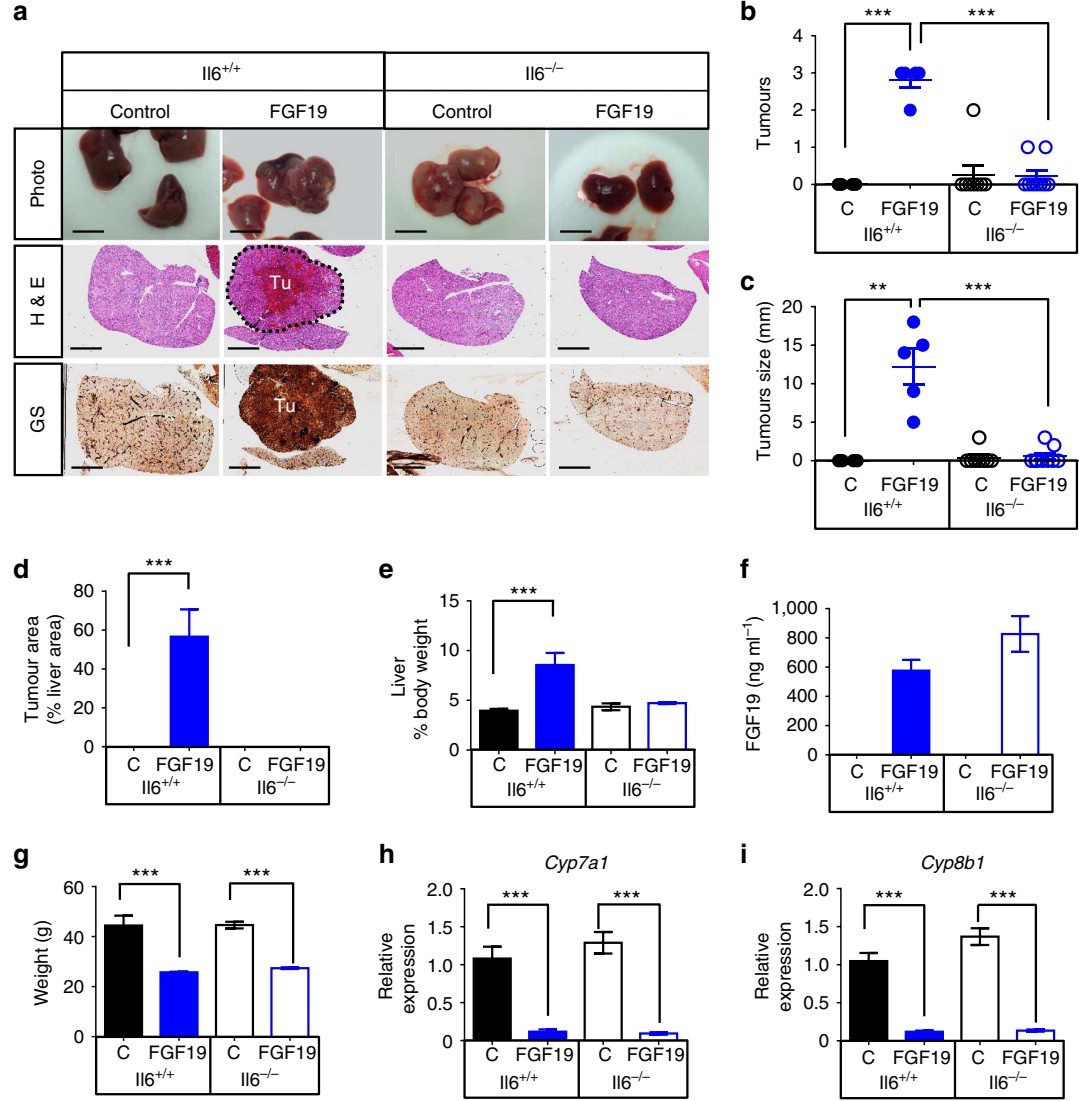

**Figure 6 | Homozygous deletion of IL-6 prevents FGF19-induced HCC development.** $Il6^{+/+}$ or $Il6^{-/-}$ mice received a single tail vein injection of AAV-*FGF19* or a control virus (C). Mice were killed after prolonged exposure to FGF19 for 12 months. (**a**) Representative images of livers from mice of the indicated genotypes are displayed. Liver tumours were assessed by H&E (middle panels) and anti-glutamine synthetase (GS, bottom panels, positive signal shown as brown colour) staining. Tu, tumours. Scale bars, 5 mm. Animal groups are $Il6^{+/+}$ control ($n=5$), $Il6^{+/+}$ FGF19 ($n=5$), $Il6^{-/-}$ control ($n=8$), $Il6^{-/-}$ FGF19 ($n=9$). (**b**) Numbers of macroscopic tumours per liver. Dots in scatterplot represent individual animals. (**c**) Tumour size. (**d**) Quantification of tumour area. (**e**) Ratios of liver-to-body weight. (**f**) Circulating FGF19 levels at the end of the study. (**g**) Body weight of the animals. (**h**) Quantitative RT-PCR of hepatic *Cyp7a1* expression. (**i**) Hepatic *Cyp8b1* mRNA levels. Values are mean ± s.e.m. ***$P < 0.001$, **$P < 0.01$ versus control group by unpaired, two-tailed *t*-test.

Taken together, these data demonstrate that disruption of IL-6/STAT3 signalling either by inhibition with an endogenous negative regulator SOCS3, a small molecule chemical JAK inhibitor, or a neutralizing antibody against IL-6, effectively abolished the pro-tumorigenic activity of FGF19 in mice.

**FGF19 correlates with STAT3 target genes in human tumours.** To assess the translational relevance and clinical significance of our findings, we systematically examined association between FGF19 expression and STAT3 activation in primary tumours from the Cancer Genome Atlas (TCGA) database, and in formalin-fixed, paraffin-embedded or frozen human HCC samples.

Our analysis of TCGA database revealed that FGF19 amplification occurs frequently in a variety of human cancers, including hepatocellular carcinomas (7%), oesophageal carcinomas (35%),

head and neck squamous cell carcinomas (24%), breast invasive carcinomas (16%) and lung squamous cell carcinomas (14%) (Supplementary Fig. 8a and Supplementary Table 1). No mutations or homozygous deletions were detected in the FGF19 in TCGA LIHC (liver hepatocellular carcinomas) data set; one hundred per cent of the genetic alterations in FGF19 detected in these tumours were identified as gene amplifications (Fig. 8a). In contrast, mutation or deletion, but not amplification, were the major types of genetic alteration detected for CTNNB1, PTEN or TP53 in human HCCs (Fig. 8a).

Next, we analysed the correlation between FGF19 gene copy number and expression level using data obtained from TCGA. FGF19 mRNA levels correlated with gene copy numbers in human liver cancer, as well as in other cancer types (Fig. 8b and Supplementary Fig. 8b). Compared with normal livers, FGF19 expression was significantly higher in HCC (Fig. 8c and

Supplementary Data 1–3). Similar elevated levels of FGF19 were observed in tumour samples and adjacent non-tumour tissues, resembling results observed in mouse models (Fig. 8c and

Supplementary Fig. 8c). Kaplan–Meier analysis showed a trend toward shorter overall survival in patients with higher FGF19 expression (Fig. 8d), consistent with prior reports[8].

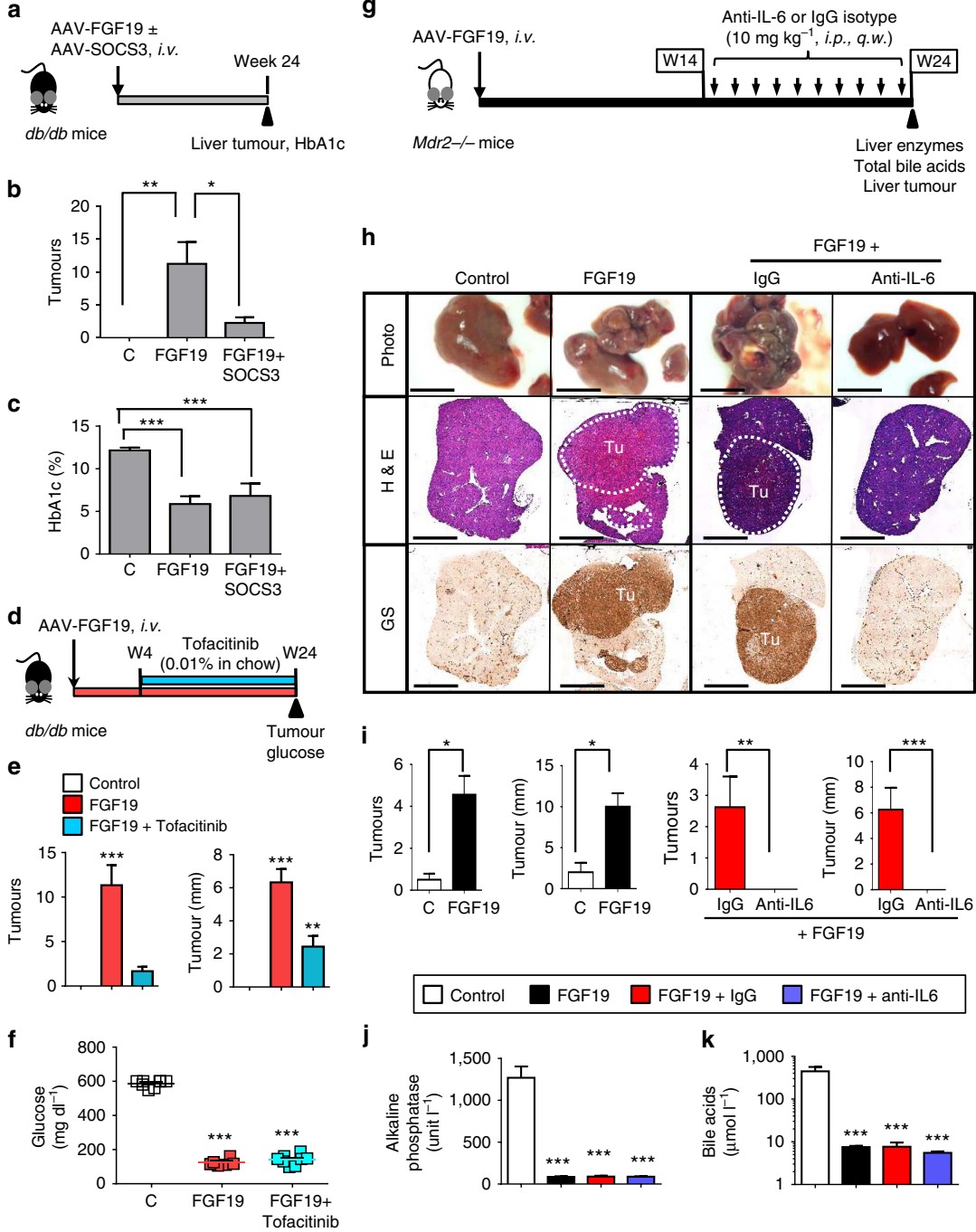

**Figure 7 | Pharmacological inhibition of the STAT3/IL-6 axis abolishes FGF19-dependent HCC.** (**a**) Schematic diagram for AAV-SOCS3 study in *db/db* mice. 11~12 week old *db/db* mice were *i.v.* administered with AAV-*FGF19* with or without AAV-SOCS3, or a control (C) virus (*n* = 5 per group). Mice were killed 24 weeks later for liver tumour analysis. (**b**) SOCS3 inhibits FGF19-induced liver tumour formation. (**c**) FGF19 normalizes HbA1c in *db/db* mice in the absence or presence of SOCS3. (**d**) Schematic diagram for tofacitinib study in *db/db* mice. 11~12-week old *db/db* mice were i.v. administered with AAV-*FGF19* or a control virus (*n* = 5 per group). Tofacitinib treatment was initiated 4 weeks later. Mice were killed 24 weeks after AAV injection. (**e**) Tofacitinib inhibits FGF19-induced liver tumour formation. (**f**) FGF19 normalizes blood glucose levels in *db/db* mice irrespective of tofacitinib treatment. Dots in scatterplot represent individual animals. (**g**) Schematic diagram for anti-IL-6 study in *Mdr2$^{-/-}$* mice. *Mdr2$^{-/-}$* mice received a single tail vein injection of AAV-*FGF19*. Starting from week 14 after AAV injection, mice were dosed intraperitoneally (*i.p.*) with 10 mg kg$^{-1}$ anti-mouse IL-6 (*n* = 10) or an isotype control antibody (*n* = 8) weekly (*q.w.*). Tumours were analysed 24 weeks after AAV administration. (**h**) Representative liver images and histological liver sections stained with H&E or anti-glutamine synthetase (brown). Tu, tumours. Scale bars, 5 mm. (**i**) Tumour multiplicity and tumour size. (**j**) Serum levels of alkaline phosphatase at the end of the study. (**k**) Serum levels of total bile acids. Values are mean ± s.e.m. ***$P$ < 0.001, **$P$ < 0.01, *$P$ < 0.05 by unpaired two-tailed *t*-test when comparing two groups, or versus control group by one-way ANOVA when comparing multiple groups.

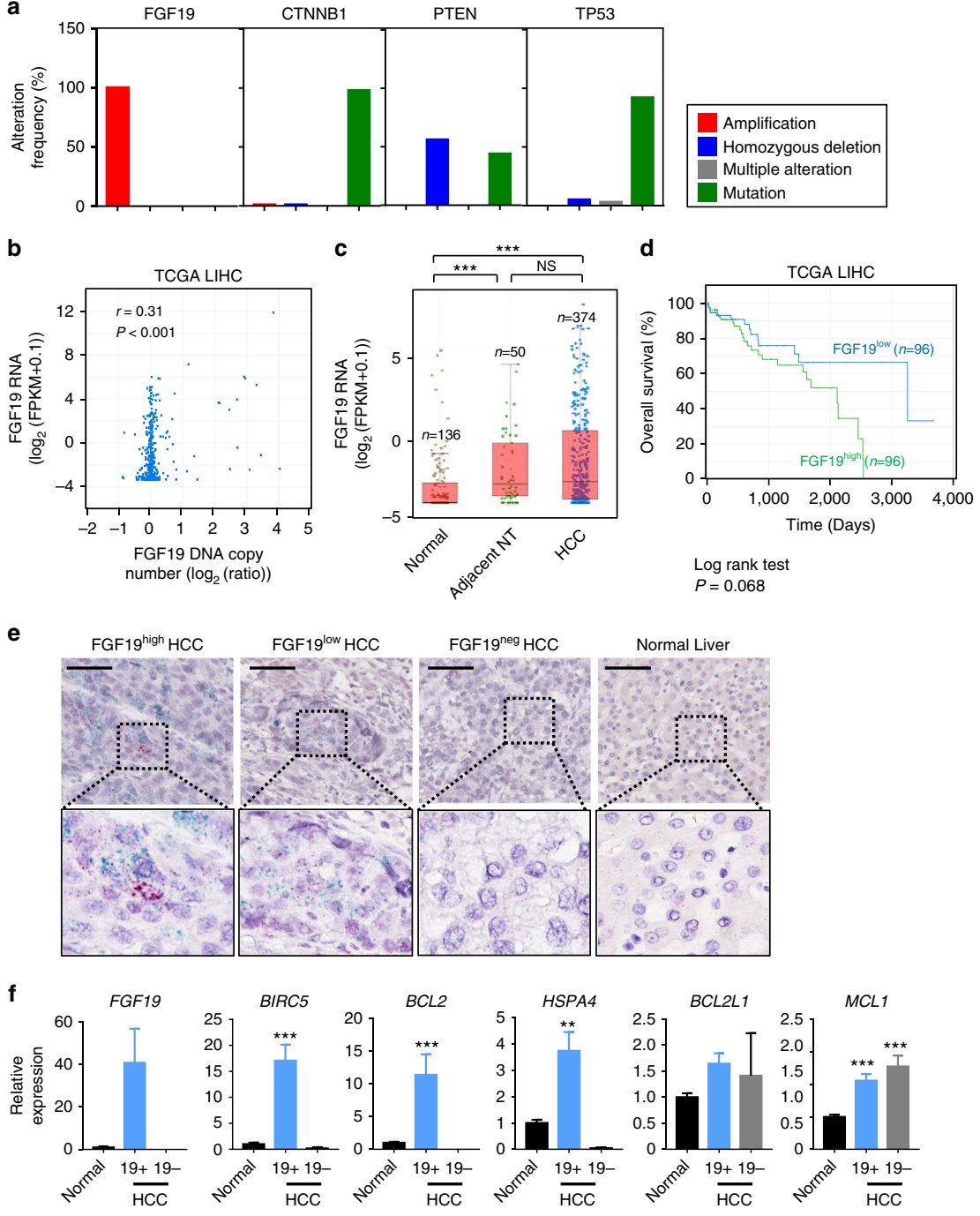

**Figure 8 | Overexpression of FGF19 and STAT3 target genes in human HCCs.** (**a**) Frequency of *FGF19* genetic alterations in human HCCs from the Cancer Genome Atlas (TCGA) liver hepatocellular carcinomas (LIHC) database. Types of alterations include amplification, homozygous deletion, mutation and multiple alterations. Genetic alterations in genes encoding CTNNB1, PTEN and TP53 are shown for comparison. (**b**) Scatter plot of 17-09742 DNA copy number versus mRNA expression in TCGA LIHC data set. Spearman correlation coefficient ($r$) and $P$ value are displayed. mRNA levels retrieved from RNA-seq data are expressed as Fragments Per Kilobase of transcript per Million mapped reads (FPKM). (**c**) Quantification of 17-09742 mRNA expression in HCC ($n = 374$), adjacent non-tumour (NT) liver tissues ($n = 50$), and normal livers ($n = 136$). Box plots depict $\log_2$ gene expression levels of 17-09742 using TCGA LIHC (for HCCs and adjacent non-tumour tissues) and GTEx (for normal livers) RNA-seq data, showing median (horizontal line) and interquartile range (box). ***$P < 0.001$ by Mann–Whitney test; NS, not statistically significant. (**d**) Kaplan–Meier survival curves of patients stratified by 17-09742 mRNA expression levels in TCGA LIHC data set. Patients were stratified into groups with high (greater than 75% rank) and low (less than 25% rank) 17-09742 expression. $P$ value from log-rank test is shown. (**e**) Co-expression of FGF19 and BIRC5, a STAT3 target gene, in human HCC samples. Duplex RNAscope chromogenic *in situ* hybridization was performed on 83 formalin-fixed, paraffin-embedded human HCC samples and 10 normal livers. Slides were co-staining with probes specific for FGF19 (red) and BIRC5 (green), and counterstained with hematoxylin. Shown are representative images from HCC samples with high, low, or negative FGF19 expression, and normal liver samples. Scale bars, 100 µm. (**f**) Upregulation of STAT3 target genes (BIRC5, BCL2, HSPA4, BCL2L1 and MCL1) in FGF19-expressing human HCCs. Quantitative RT-PCR was performed on RNA extracted from frozen FGF19-expressing (19 + ) human HCC samples ($n = 5$), FGF19-non-expressing (19 − ) human HCC samples ($n = 5$), or normal livers ($n = 5$). Values are mean ± s.e.m. ***$P < 0.001$, **$P < 0.01$ by one-way ANOVA.

Further analysis of TCGA database revealed that FGF19 expression is correlated with mRNA levels of STAT3 target genes, such as *BIRC5, BCL2, HSPA4* and *BCL2L1* (Supplementary Fig. 8d). In addition, we used *in situ* hybridization to measure FGF19 and BIRC5 mRNA levels in 83 human HCC specimens and 10 normal livers (Fig. 8e, Supplementary Data 4 and Supplementary Table 2). Notably, FGF19 mRNA was undetectable using this technique in any of the normal liver tissues. In contrast, FGF19 mRNA levels were significantly elevated in ten of the formalin-fixed, paraffin-embedded human HCC specimens tested (12%). Moreover, FGF19 and BIRC5 were concomitantly overexpressed in all 10 of these tumour samples, further establishing a positive correlation between the expression of FGF19 and *BIRC5*, a STAT3 target gene, in human HCC. Upregulation of STAT3 target genes, including *BIRC5, BCL2, HSPA4*, was also confirmed by quantitative RT-PCR analysis in frozen human HCC specimens expressing FGF19, but not in HCC samples lacking FGF19 expression (Fig. 8f and Supplementary Table 3).

Collectively, these data support the notion that activation of the STAT3 pathway by FGF19 may be clinically relevant in human HCC pathogenesis and prognosis.

## Discussion

Patients with HCC have experienced little improvement in overall survival in the past decades. Amplification of *FGF19* is a recurring theme in human HCC, however, the signalling pathways leading to FGF19-induced hepatocellular carcinogenesis still remain to be elucidated. Using genetic and pharmacological tools, we have uncovered a previously unsuspected role for cytokine IL-6 and its downstream effector STAT3 in *FGF19*-associated tumorigenesis, as well as providing evidence that the liver microenvironment can be modified to limit tumorigenesis in response to FGF19.

Our results showed that hepatocellular STAT3 is a critical effector in FGF19-induced HCC formation. STAT3 has a complex but well-established role in coordinating pro-tumorigenic gene expression and immunomodulatory functions, and aberrant activation of STAT3 has been reported in a wide variety of haematologic and epithelial tumours[20,21]. Although mutations in the *STAT3* gene are rarely identified, persistent activation of the STAT3 pathway is a common phenotype in human HCC[5]. In defining a central role for STAT3 in FGF19-driven hepatocarcinogenesis, the data presented in this report extend previous findings on the causal role played by STAT3 in hepatocellular tumorigenesis[44,45], and suggest that STAT3 may serve as a common signalling node in the development of HCC associated with diverse etiologies, including viral infection, and metabolic and hormonal dysregulation.

Activated by members of the IL-6 family of cytokines, STAT3 signalling is mediated through a receptor complex comprising the ubiquitously expressed gp130 receptor and a set of ligand-specific receptors on specific target cells[39]. In this report, we show that among members of gp130 ligand family, IL-6 is uniquely induced by FGF19 and is essential for FGF19-mediated cellular transformation and growth *via* STAT3 phosphorylation. IL-6 is a multifunctional cytokine important for immune responses, cell survival, apoptosis and proliferation[39]. Elevated levels of IL-6 have been associated with HCC in human patients[46], and in the case of Castleman's disease, hypersecretion of IL-6 leads to the formation of HCC tumours, as well as systemic inflammatory symptoms[47]. In preclinical models, transgenic mice expressing IL-6 and its receptor IL-6Rα develop hepatocellular hyperplasia and hepatocellular adenomas[48]. More recently, IL-6 signalling has been identified as a key pathway promoting the expansion of liver cancer progenitors and/or cancer stem cells[49,50]. These liver tumour-initiating cells represent a rare population of liver cells and express surface markers such as CD24 (ref. 51), CD90 (ref. 52), CD133 (ref. 53) or EpCAM (ref. 54). Although we did not observe a proliferative effect of IL-6 (or in combination with FGF19) on cultured hepatocytes, further studies are needed to examine direct effect of IL-6 on enriched hepatocyte subpopulations expressing these stem cell markers.

Physiologically, FGF19 signalling *via* FGFR4 suppresses *de novo* bile acid production in the liver, tightly maintaining hepatic and systemic levels of these detergent-like molecules at a physiological threshold. Under pathological conditions, marked elevation in hepatic and plasma levels of FGF19 were observed in patients with HCC from diverse etiologies, and in patients with extrahepatic cholestasis due to biliary obstruction[8,9,55]. In this context, upregulation of FGF19 either through amplification or overexpression may represent an adaptive response to limit hepatic bile acid accumulation and tissue damage in response to acute and chronic liver diseases, and provide a survival advantage under cholestatic conditions. This beneficial host response may, however, come at the expense of cell proliferation, tumour initiation and progression associated with prolonged exposure to elevated FGF19 levels.

Taken together, our studies support a model whereby FGF19 not only acts directly on hepatocytes to limit *de novo* bile acid synthesis, but also *via* a non-cell-autonomous signalling pathway to influence the hepatic microenvironment comprising a variety of cell types, including residential and infiltrating immune cells. In the cholestatic liver, FGF19-mediated repression of *Cyp7a1* limits bile acid accumulation, whereas IL-6 secretion from hepatic immune cells may protect hepatocytes from pathogens. In any case, given that the hepatocyte is the only cell type that co-expresses FGFR4 and KLB in the liver, the precise mechanism by which FGF19 promotes IL-6 secretion from innate immune cells remains to be determined.

Our study does not exclude other mechanisms by which FGF19 may promote tumorigenesis, and the non-cell-autonomous signalling described in this report could work in parallel with such pathways. For example, activation of the WNT/β-catenin signalling by FGF19 has been reported in cancer cell lines[56], potentially through a mechanism involving crosstalk with the EGFR pathway[57]. FGFR4, for which FGF19 is a high-affinity ligand, has been identified as a marker for cancer prognosis and disease progression[58,59]. Recent studies revealed that substitution of the conserved glycine 388 residue in FGFR4 with a charged arginine residue, resulting from a commonly observed single-nucleotide polymorphism, alters the transmembrane spanning segment in the receptor and exposes a membrane-proximal STAT3 binding site[60]. Although this report focuses on the non-cell-autonomous mechanism of FGF19-induced hepatocarcinogenesis, contributions from cell-autonomous mechanisms cannot be ruled out, for example, in hepatocytes that acquire the ability during malignant transformation to produce IL-6.

Our analysis of the TCGA database revealed relatively high frequencies of *FGF19* gene amplification in a variety of tumours. Moreover, *FGF19* mRNA levels correlate with the expression of STAT3 target genes, such as BIRC5, BCL2, HSPA4 and BCL2L1. In a previous report, Hyeon and colleagues screened 281 human HCC samples and found that increased expression of *FGF19* correlates with poor prognosis in these patients[8]. Therefore, our findings in mice regarding the activation of the STAT3 pathway by FGF19 may have translational relevance in human HCC pathogenesis and prognosis, although the extent to which our observations in mice translate to human remains unclear and warrants further investigation.

With the availability of genomic profiling data derived from large numbers of patients, our understanding of the genetic

landscape in cancer has significantly improved. However, challenges remain in translating this knowledge base into therapeutic benefits for patients. On the basis of the data we have presented in this report, the selective inhibition of IL-6/STAT3 signalling in the tumour microenvironment provides an attractive target for therapeutic intervention. Notably, IL-6 appears to be an essential factor common to the induction of HCC by a variety of stimuli, including chemical[61], obesity[62] and FGF19. Moreover, given that *FGF19* is also amplified or over-expressed in oesophageal, lung, breast and colon cancers, the mechanism by which FGF19 promotes tumorigenesis through the IL-6/STAT3 cascade may be more broadly applicable to multiple tumour types, and the pharmacological treatment strategy proposed here might be applicable to a wide spectrum of patients whose tumours rely on the constitutive engagement of this pathway.

## Methods

**Animals and animal care.** All experimental procedures were approved by the Institutional Animal Care and Use Committee at NGM. Mice were housed in a pathogen-free animal facility at 22 °C under controlled 12-h light and 12-h dark cycles. All mice were maintained in filter-topped cages on standard chow diet (Teklad 2918) or diets containing inhibitors when indicated, and autoclaved water *ad libitum*. Male mice were used unless otherwise specified. Sample sizes were determined on the basis of homogeneity and consistency of characteristics in the selected models and were sufficient to detect statistically significant differences in tumorigenicity and metabolic parameters between groups. Mice were randomized into the treatment groups based on body weight and blood glucose. Studies were replicated in 2–3 independent cohorts of animals. All injections and tests were performed during the light cycle. Investigators were not blinded during these studies. No animals were excluded from analysis at study completion. *db/db* mice (BKS.Cg-*Dock7*$^{m+/+}$ *Lepr*$^{db}$/J, #000642), *Stat3*$^{f/f}$ mice (B6.129S1-*Stat3*$^{tm1Xyfu}$/J, #016923), *Il6*$^{-/-}$ mice (B6.129S2-*Il6*$^{tm1Kopf}$/J, #002650), *Mdr2*$^{-/-}$ mice (FVB.129P2-*Abcb4*$^{tm1Bor}$/J, #002539) and wild type control C57BL6/J (#000664) mice were obtained from Jackson Laboratory.

**Bioinformatics analysis.** The TCGA data on human tumour samples (http://gdc-portal.nci.nih.gov) and GTEx data on normal human livers (http://www.gtexportal.org) referenced in this study are available in public repositories. The patient clinical and pathological features are summarized in Supplementary Tables 1 and Supplementary Data 1,2 (TCGA database), and Supplementary Data 3 (GTEx database). Bioinformatics data analysis, including detection of amplification, deletion or mutations, was conducted using ArrayStudio software version 9.0 from OmicSoft. Correlations between gene copy number and the corresponding gene expression data, as well as correlations of FGF19 and STAT3 target gene expression, were determined by Spearman rank order correlations test. Kaplan–Meier analysis was used to compare patient survival by the log rank test.

**Patient samples.** Pathologically confirmed human HCC samples were collected under informed consent after approval by the Institutional Review Board, and procured by US BioMax. All formalin-fixed, paraffin-embedded (FFPE) specimens, including 83 HCC tumour samples and 10 normal liver tissues, were tested negative for HIV. Frozen FGF19-expressing and FGF19-non-expressing human HCC specimens (CrownBio) and frozen normal liver samples (BioReclamation/IVT) were tested negative for HIV and HCV. Detailed information on age, gender and tumour grade/stage/type is shown in Supplementary Data 4 and Supplementary Tables 2–3.

**RNAscope duplex *in situ* hybridization.** RNA *in situ* hybridization was performed using the RNAscope 2.5 HD duplex chromogenic assay (Advanced Cell Diagnostics) according to manufacturer' protocol. Briefly, 5 µm formalin-fixed, paraffin-embedded tissue sections were baked for 1 h in an oven at 60 °C. After de-paraffinization, target retrieval was performed at 95 °C for 15 min, and slides were incubated with proteases at 40 °C for 30 min. Human FGF19 probe (#553981-C2, 20 double Z probe pairs against region 451-2128nt in NM_005117.2) and BIRC5 probe (#465361, 20 double Z probe pairs against region 834-1868nt in NM_001168.2) were incubated for 2 h at 40 °C before signal amplification and detection, and counterstaining with hematoxylin. Slides were scanned using Aperio image software.

**DNA constructs.** Human FGF19 cDNA (NM005117) was sub-cloned into pAAV-EF1α vector using SpeI and NotI sites with primers 5′- CCGACTAGTCACCatgcg gagcgggtgtgtgg -3′ (sense) and 5′- ATAAGAATGCGGCCGCTTACTTCTCAA

AGCTGGGACTCCTC-3′ (antisense). cDNAs for TBG promoter[22] and Cre recombinase (AF298789) were chemically synthesized (DNA2.1), and sub-cloned into promoter-less pAAV vector.

**Cell culture.** All cells were cultured in a humidified incubator with 5% $CO_2$ and 95% air at 37 °C. Cell lines used were confirmed to be mycoplasma free and authenticated by short tandem repeat DNA profiling.

**AAV production.** AAV293 cells (Agilent Technologies) were cultured in Dulbeco's Modification of Eagle's Medium (DMEM; Mediatech) supplemented with 10% fetal bovine serum (FBS) and 1 × antibiotic-antimycotic solution (Mediatech). The cells were transfected with three plasmids (AAV transgene, pHelper (Agilent Technologies) and AAV2/9) for viral production. Viral particles were purified using a discontinued iodixanal (Sigma) gradient and re-suspended in phosphate-buffered saline (PBS) with 10% glycerol and stored at − 80 °C. Viral titre or vector genome number was determined by quantitative PCR using custom Taqman assays specific for Woodchuck hepatitis virus posttranscriptional regulatory element (WPRE) sequences. Standard curves for WPRE were obtained from serial dilutions over a six log range of the corresponding plasmids. AAV-mediated gene delivery provides a means to achieve long-lasting transgene expression without the inflammatory responses that are commonly associated with other viral vectors. When introduced into adult mice, sustained expression of up to one year has been observed.

**Hepatocyte-specific Stat3 deletion.** For studies in *Stat3*$^{f/f}$ mice (B6.129S1-*Stat3*$^{tm1Xyfu}$/J, Jackson Laboratory, #016923), 14–18 week-old mice received a single intravenous dose of $1 \times 10^{11}$ v.g. of AAV-*FGF19* in combination with $3 \times 10^{11}$ v.g. of AAV-TBG-Cre recombinase or AAV-GFP through the tail vein. *Stat3*$^{f/f}$ mice injected with AAV-TBG-Cre alone served as *Stat3*$^{\Delta Hep}$ controls. *Stat3*$^{f/f}$ mice were used as wild type controls. AAV-TBG-Cre drives Cre recombinase expression under TBG promoter, which allows hepatocyte-specific expression. After 12 months AAV administration, mice were killed and examined for liver tumour formation. The maximum diameter of liver tumour nodules in each mouse was measured with a caliper and total numbers of tumour nodules per liver were recorded. Livers were weighed and collected for histological examination or gene expression analysis.

**Glucose and body composition.** Blood concentrations of *ad libitum* fed glucose were measured in conscious animals from a hand-held glucometer (Accu-check, Roche Diagnostics) using tail vein blood. For body composition measurements, un-anaesthetised mice were placed in 'live' probe, and scanned for 1 min inside the EcoMRI-5000 whole body composition analyzer. Fat mass and lean mass were calculated using EcoMRI software from multiple primary accumulation numbers.

**Gene expression analysis.** Tissues were snap-frozen in liquid nitrogen upon killing of animals. Total RNA was extracted using RNeasy Mini kit (Qiagen) and treated with DNase I (Thermo Fisher Scientific). Real-time quantitative PCR (qRT-PCR) assays were performed using QuantiTect multiplex qRT-PCR master mix (Qiagen) and premade Taqman gene expression assays (Life Technologies). Samples were loaded into an optical 384-well plate and qRT-PCR were performed in duplicates on QuantStudio 7 Flex Real-Time PCR System (Applied Biosystems). After an initial hold at 50 °C for 30 min to allow reverse transcription to complete, HotStart Taq DNA polymerase was activated at 95 °C for 15 min. Forty cycles of a three-step PCR (94 °C for 45 s, 56 °C for 45 s and 76 °C for 45 s) were applied and the fluorescence intensity was measured at each change of temperature to monitor amplification. Target gene expression was determined using the comparative threshold cycle (ΔΔCt) method and normalized to the expression of housekeeping genes glyceraldehyde 3-phosphate dehydrogenase (GAPDH) or β-actin.

**Blood parameters.** Blood was collected from tail vein in un-anaesthetised animals using microvette serum gel tubes (Sarstedt) for measurements of insulin, ALP, ALT, AST, bile acids and FGF19 concentrations. Serum samples were prepared by centrifugation at 4 °C for 10 min at 6,000 g after clotting at room temperature for 30 min. For HbA1c measurements, whole blood was collected into EDTA-tubes (Becton Dickinson) after euthanasia by cardiac puncture and the tubes were quickly inverted multiple times to prevent clotting. HbA1c (from whole blood samples) and liver enzyme (from serum samples) levels were measured on Cobas Integra 400 Plus Clinical Analyzer (Roche Diagnostics). Serum FGF19 level was determined by FGF19 enzyme-linked immunosorbent assays (ELISA) (Biovendor, RD191107200R). Serum insulin was assessed with ELISA kits from ALPCO (80-INSMS-E10). Concentrations of total bile acids in serum were determined using a 3β-hydroxysteroid dehydrogenase method (Diazyme, DZ042A). All assays were performed according to the manufacturers' instructions.

**Histology and immunohistochemistry.** Mouse livers were fixed in 10% neutral-buffered formalin and embedded in paraffin. 5 µm sections were deparaffinized in xylenes (5 min), rehydrated sequentially in graded ethanol (100, 95, 80, 70, 50%, 2 min each) and PBS (2 min). Hematoxylin and eosin staining was performed using

standard methods. For immunohistochemistry, specimens were subjected to antigen retrieval in a citrate-based Antigen Unmasking Solution (Vector Laboratories, H-3300), and incubated for 30 min with 3% $H_2O_2$ at room temperature to block endogenous peroxidase activity. Sections were blocked in PBST (PBS + 0.1% Tween-20) containing 10% goat serum, stained with primary antibodies against BrdU (Dako), glutamine synthetase (Abcam), PCNA (Dako), Ki-67 (Dako) or STAT3 (Cell Signaling Technology) diluted in blocking solution at 4 °C overnight (see Supplementary Table 4 for detailed information on antibodies used). Specimens were then washed three times for 5 min each in PBST and incubated with biotinylated secondary antibodies (Supplementary Table 4) in blocking solution for 1 h at room temperature. R.T.U. Vectastain ABC peroxidase reagent (Vector Laboratories, PK-7100) and di-amine-benzidine (DAB) colorimetric peroxidase substrate (Vector Laboratories, SK-4100) were used for detection. For immuno-histochemical staining with anti-BrdU antibody, DNA was denatured in liver sections with 0.1 N HCl (37 °C for 5 min), followed by neutralization with 0.1 M sodium borate buffer (pH 8.5, 5 min). Sections were further treated with DNase I (GE Healthcare) before incubation with anti-BrdU antibodies. When indicated, sections were counterstained with hematoxylin. Digital imaging microscopy was performed using a Leica DM4000 microscope equipped with DFC500 camera and a high-precision motorized scanning platform (Leica). Images for the entire liver section was acquired by Turboscan and real-time imaging stitching at camera frame rates using Surveyor program. For morphometric analysis of tumour area, glutamine synthetase-positive tumour areas were quantified using Measure/Count/Area tool from ImagePro software. Ki-67 labelling index is calculated as percentage of Ki-67-positive nuclei among total nuclei in the area scored. A minimum of 1,000 cells from 5 randomly-selected fields were counted.

**In vivo pSTAT3 activation.** Production of recombinant FGF19 protein has been described previously[12]. Saline solution (0.9% NaCl) is used as vehicle for dosing in mice. 11–12-week old *db/db* mice (BKS.Cg-Dock7$^{m +/+}$ Lepr$^{db}$/J, #000642) were injected intraperitoneally with 1 mg kg$^{-1}$ FGF19 protein. Mice were killed 2 h later for serum and liver collection. Liver samples were homogenized in Tissue Extraction Reagent I (ThermoFisher Scientific) supplemented with Complete Mini protease inhibitor cocktail (Roche) and phosphatase inhibitor cocktail (Sigma). For IL-6 inhibition, mice were intraperitoneally injected with 10 mg kg$^{-1}$ anti-IL-6 blocking antibody (*InVivo*Mab clone MP5-20F3, Bio-X-Cell) or an isotype control IgG1 antibody (*InVivo*Mab clone HRPN, Bio-X-Cell). 10 min later, mice were dosed with 1 mg kg$^{-1}$ FGF19 protein. Livers were collected 2 h after FGF19 injection for pSTAT3 analysis by immunoblotting.

**Isolation of primary hepatocytes and non-parenchymal cells.** To isolate primary mouse hepatocytes, livers were perfused with 30 ml Liver Perfusion Media (Invitrogen) followed by 50 ml Liver Digest Media (Invitrogen) at 37 °C using a peristaltic pump (3 ml min$^{-1}$, GE Healthcare). Liver capsule membrane was peeled back with a forceps and liver cells were resuspended in DMEM/5%FBS and passed through a 70 μm cell strainer (Becton Dickinson). Hepatocytes were pelleted by low-speed centrifugation at 50 g for 3 min and washed twice with DMEM/5% FBS. Supernatants from the repeated low-speed centrifugation were collected and non-parenchymal cells were pelleted by centrifugation at 500g for 10 min. Fresh human hepatocytes were isolated from whole livers of consented donors the same day they were received using a collagenase digestion method (BioreclamationIVT).

**In vitro pSTAT3 activation.** Primary mouse hepatocytes were diluted in attach-ment media (DMEM supplemented with 2 mM glutamine, 100 unit ml$^{-1}$ penicillin, 100 unit ml$^{-1}$ streptomycin, and 5% FBS), and 1.5 × 10$^5$ cells were seeded on Biocoat collagen-coated 6-well plate (Becton Dickinson) in 2 ml attachment media. After 4 h, cells were changed into M199 media (Invitrogen) containing 100 nM dexamethasone, 100 nM triido-L-thyronine (T3) and 1 nM insulin, and cultured for 16 h at 37 °C. When indicated, 50 ng ml$^{-1}$ FGF19 protein or 50 ng ml$^{-1}$ mouse IL-6 (Peprotech) were added, and cells were lysed 5, 15, 30, 60 or 120 min after ligand addition.

Primary human hepatocytes were diluted in InvitroGRO CP media containing Torpedo antibiotics (BioreclamationIVT), and 5 × 10$^5$ cells were seeded on collagen-coated 6-well plate. Four hours later, cells were changed into William's E media (Invitrogen) containing 100 nM dexamethasone, 1x ITSG supplement (Invitrogen), 0.25 mg ml$^{-1}$ Matrigel (Becton Dickinson), and cultured for 16 h at 37 °C. When indicated, 50 ng ml$^{-1}$ FGF19 protein or 50 ng ml$^{-1}$ human IL-6 (Peprotech) were added, and cells were lysed 5, 15, 30, 60 or 120 min after ligand addition.

Cell extracts were prepared in 50 mM Tris-Cl, PH 7.2, 150 mM NaCl, 1% NP-40, 0.1% SDS, 0.5% sodium deoxycholate supplemented with a protease inhibitor cocktail (Roche) and phosphase inhibitors (Sigma, phosphatase inhibitor cocktail set 2 and set 3), and protein concentrations were determined by BCA protein assay (Thermo Scientific).

**Immunoblotting.** Cell or tissue lysates were mixed with Laemmli sample loading buffer, heated for 5 min at 95 °C, and separated on Criterion TGX 4–20% pre-cast gels (Bio-Rad) in 1 × MOPS running buffer. Proteins were transferred onto nitrocellulose membranes on Trans-Blot Turbo Transfer System (Bio-Rad).

Membranes were blocked in TBST buffer (20 mM Tris–Cl, pH 7.6, 137 mM NaCl, 0.1% Tween-20) containing 5% non-fat dry milk for 1 h at room temperature, and incubated with anti-β-actin (Sigma), anti-ERK1/2 (Santa Cruz), anti-pERK1/2$^{T202Y204}$, anti-pSTAT1$^{Y701}$, anti-pSTAT3$^{Y705}$, anti-pSTAT5$^{Y694}$, pSTAT6$^{Y641}$ or anti-STAT3 antibodies (see Supplementary Table 4 for detailed information on antibodies) in blocking buffer at 4 °C overnight. Membranes were washed in TBST extensively, and bound proteins were detected with horseradish peroxidase-conjugated goat-anti-mouse IgG or goat-anti-rabbit IgG antibodies (GE Healthcare) for one hour at room temperature. Signals were developed with SuperSignal West Dura extended duration substrate (Thermo Scientific), and captured with a ChemiDoc imaging system (Bio-Rad).

**In vivo BrdU incorporation.** All surgical procedures in mice were performed under isoflurane inhalation anaesthesia (2–3%). The hair over the intrascapular area was shaved and skin aseptically prepared. A small incision was made using a blunt tipped scissor, and a small pocket formed by spreading the subcutaneous connective tissue apart. Micro-osmotic pumps (Alzet, 1007D) releasing 8.5 mg kg$^{-1}$ day$^{-1}$ 5-bromo-2-dexoyuridine (BrdU, Sigma) and 0.4 mg kg$^{-1}$ day$^{-1}$ FGF19 protein were implanted subcutaneously into the pocket with the flow moderator pointing away from the incision on day 1. Incisions were closed with 3-0 silk suture (Ethicon) and wound clips. Mice were killed on day 6, and serum and livers were collected for exposure and BrdU incorporation analysis. Saline solution (0.9% sodium chloride) was used as vehicle for dilutions.

**In vitro BrdU incorporation.** Primary mouse hepatocytes were diluted in attachment media (DMEM supplemented with 2 mM glutamine, 100 unit ml$^{-1}$ penicillin, 100 unit ml$^{-1}$ streptomycin, and 5% FBS), and 1.5 × 10$^4$ cells per well in 100 μl were seeded on a clear-bottom, black-wall collagen-coated 96-well plate (Becton Dickinson). Four hours later, cells were changed into M199 media (Invitrogen) containing 100 nM dexamethasone, 100 nM triido-L-thyronine (T3), and 1 nM insulin, and cultured for 16 h at 37 °C. Various concentrations of FGF19 or mouse HGF (Peprotech) proteins were added, followed by pulsing with 10 μl per well BrdU Labelling Solution (Roche Diagnostics) 24 h after ligand stimulation. Cells were fixed 48 h after ligand addition, and chemiluminescent BrdU signals were detected per manufacturer's instruction (Roche Diagnostics, 11 669 915 001).

Primary human hepatocytes were diluted in InvitroGRO CP media containing Torpedo antibiotics (BioreclamationIVT), and 4.5 × 10$^4$ cells were seeded on a clear-bottom, black-wall collagen-coated 96-well plate. Four hours later, cells were changed into William's E media (Invitrogen) containing 100 nM dexamethasone, 1 × ITSG supplement (Invitrogen), 0.25 mg ml$^{-1}$ Matrigel (Becton Dickinson), and cultured for 16 h at 37 °C. Various concentrations of FGF19 or human HGF (Peprotech) proteins were added, followed by pulsing with 10 μl per well BrdU Labelling Solution 24 h after ligand stimulation. Cells were fixed 48 h after ligand addition, and chemiluminescent BrdU signals were detected per manufactor's instruction (Roche Diagnostics, 11 669 915 001).

**Flow cytometry.** For quantification of in vivo BrdU incorporation by flow cytometry, isolated hepatocytes were resuspended in Cytofix/Cytoperm buffer (Becton Dickinson), washed with Perm/Wash buffer, and treated with DNase at 37 °C for one hour to expose incorporated BrdU. Cells were subsequently stained with anti-BrdU-APC (Becton Dickinson, 552598) for 30 min at room temperature, and washed extensively with Perm/Wash buffer. For two-colour flow cytometry analysis, cells were resuspended in staining buffer (PBS + 3% FBS + 0.1% sodium azide) containing 7-AAD and run at a rate no greater than 300 events/second on a FACSCalibur instrument. Fluorescent data were collected using CellQuest software and analysed using Flowjo program.

For intracellular IL-6 staining, isolated hepatic non-parenchymal cells were incubated in RPMI1640 media containing 50 ng ml$^{-1}$ FGF19, 10% FBS, 100 unit ml$^{-1}$ penicillin, 100 unit ml$^{-1}$ streptomycin, and protein transport inhibitors (10 μM brefeldin A and 2 mM monensin, eBioscience) at 37 °C for 4 h. Cells were washed and resuspended in staining buffer, and non-specific binding was blocked by Mouse Fc Blocker (2.4G2, eBioscience, 1:100) for 30 min on ice. Cells were subsequently stained with fluorophore-labelled antibodies (CD45, clone 30 F-11; CD3, clone 17A2; CD19, clone 6D5; CD11b, clone M1/70; Ly6-G, clone 1A8; NK1.1, clone PK136; F4/80, clone BM8) or isotype-control antibodies for 30 min at 4 °C in the dark (see Supplementary Table 5 for detailed information on antibodies). Live/Dead-DAPI kit (Invitrogen) was used to gate out dead cells. Surface-stained cells were fixed and permeabilized using Fix/Perm Buffer set (eBioscience), and incubated with anti-IL-6-APC (clone MP5-20F3) in 1xPerm buffer or isotype control antibody for 60 min at 4 °C in the dark. Cells were analysed on a LSRFortessa X-20 flow cytometer configured with 5 lasers (blue, red, violet, green and ultraviolet) to detect up to 16 parameters in multicolour flow cytometry. Compensation setup for multi-colour flow cytometric analysis was conducted using CompBead (BD Biosciences). Specifically, CompBead particles were mixed with fluorochrome-conjugated antibodies to provide distinct positive and negative stained populations, and were then used to set compensation levels with the automated compensation setup available in FACSDiva software. Data were collected using FACSDiva software and analysed using Flowjo programs.

**Tumorigenicity study in IL-6-deficient mice.** $Il6^{-/-}$ mice (B6.129S2-$Il6^{tm1Kopf}$/J, Jackson Laboratory, #002650) were backcrossed to C57BL/6 mice for at least nine generations. Age and gender-matched wild type C57BL6 mice (Jackson Laboratory, #000664) were used as $Il6^{+/+}$ controls. $1 \times 10^{11}$ v.g. of AAV-FGF19 or AAV-GFP were injected intravenously in a volume of 200 µl saline to 3–4 month-old mice. Body weight, body composition, and blood glucose were measured at designated time. Mice were killed 12 months post AAV administration for liver tumour assessment.

**Tumorigenicity study in db/db mice.** 11–12 week-old db/db mice received a single intravenous dose of AAV containing either FGF19 or a control gene GFP. For AAV-SOCS3 inhibition, mice were co-injected with AAV-FGF19 ($5 \times 10^9$ v.g.) and AAV-SOCS3 ($5 \times 10^{10}$ v.g.). For tofacitinib treatment, chow diet containing 0.01% (w/w) tofacitinib was started 4 weeks post AAV-FGF19 ($1 \times 10^{11}$ v.g.) injection, and continued ad libitum for 20 additional weeks. Mice were killed 24 weeks post AAV administration and examined for liver tumour formation by gross appearance and histology. Body weight and blood glucose were monitored at designated times.

**Anti-IL-6 treatment in $Mdr2^{-/-}$ mice.** 4 month-old female $Mdr2^{-/-}$ mice received a single intravenous dose of $1 \times 10^{11}$ v.g. of AAV-FGF19 or a control virus AAV-GFP. We started treating AAV-FGF19-injected $Mdr2^{-/-}$ mice with anti-IL-6 14 weeks after AAV administration. Mice were dosed with anti-mouse IL-6 (InVivoMab clone MP5-20F3, Bio-X-Cell) or an isotype control IgG1 antibody (InVivoMab clone HRPN, Bio-X-Cell) once weekly i.p. at 10 mg kg$^{-1}$ for a total of 10 weeks. Mice were killed 24 weeks post AAV administration for liver tumour assessment. Serum levels of liver enzymes (ALP, ALT, AST) and bile acids were measured at indicated time points.

**Immune cell depletion.** Kupffer cells were depleted by clondronate liposome (i.v., 200 µl per mouse every three days, for a total of three doses). PBS liposomes were used as controls. To deplete neutrophils or CD8+ T cells, 11-12 week old db/db mice were injected i.p. with 500 µg anti-mouse Ly-6G antibody (clone 1A8, Bio-X-Cell) or anti-mouse CD8 antibody (clone 53-6.72, Bio-X-Cell), respectively.

**Data analysis.** in vitro experiments were performed in triplicates and repeated at least twice. in vivo experiments were conducted with cohorts of 3–9 mice per group (n values detailed in figure legends), with individual mouse data shown when indicated. All results are expressed as mean ± s.e.m. Unpaired two-tailed Student's t-test was used to compare two treatment groups (GraphPad Prism). One-way ANOVA followed by Dunnett's post-test was used to compare data from multiple groups (GraphPad Prism). A P value of 0.05 or less was considered statistically significant.

**Data availability.** The TCGA and GTEX data referenced in the study are available in public repositories (http://gdc-portal.nci.nih.gov for TCGA, and http://www.gtexportal.org for GTEX). All other data supporting the findings of this study are available within the article and its Supplementary Information files.

Uncropped immunoblots accompanied with locations of molecular weight markers (Supplementary Fig. 9) and additional experimental details are available in the Supplementary Information.

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

## Acknowledgements

We thank Dr Josh Lichtman for bioinformatics support. We thank Drs Jin-Long Chen, Alex DePaoli, Steve Rossi and Husam Younis for advice and insightful discussions. We thank Xueyan Wang, Danielle Holland, Xunshan Ding, Van Phung and Mark Gilbert for technical assistance, and NGM vivarium staff for the care of the animals used in the studies. Funding was provided by NGM Biopharmaceuticals, Inc. All authors are employees and stockholders of NGM Biopharmaceuticals, Inc.

## Author contributions

Study design: H.T. and L.L.; acquisition of data: M.Z., H.Y. and L.L.; analysis and interpretation of data: M.Z., R.M.L., H.T. and L.L.; drafting of the manuscript: R.M.L. and L.L.

## Additional information

**Competing interests:** All authors are employees and stockholders of NGM Biopharmaceuticals, Inc.

