## [Peer Review File · Nature Communications]

Reviewers' comments:

Reviewer #1 (Remarks to the Author):

In the present manuscript the authors aim to characterize the mechanism of FGF19-dependent tumor initiation through a non-cell-autonomous activation of IL6/STAT3 signaling pathway. The study adds information regarding the mechanism of FGF19-induced hepatocarcinogenesis, particularly in relation to JAK/STAT3 signaling activation. Nonetheless several issues remain to be clarified.

Major comments:

- 1) Human data: The authors state that FGF19 overexpression would lead to STAT3 activation in HCC. From a translational perspective, this hypothesis requires to be confirmed in human samples.
- 2) Mechanism of induction of IL6 overexpression. The authors state that FGF19 overexpression induces release of IL6 from the innate immune cells from the microenvironment. The authors need to clarify if this is occurring in the microenvironment of the tumor or just of the adjacent tissue. This reviewer is concerned by the fact that the information about IL6 overexpression in the tumor is lacking.
- 3) Is tumor proliferation related to STAT 3 activation?. The study is focused on activation of STAT3 signaling as the main driver of FGF19 -induced tumorigenesis. However, the model describes activation of WNT/beta-catenin signaling, and translocation of this protein into the nucleus. Can the authors explore if this might be also a mechanism of proliferation, and whether there is a direct correlation with FGF19 levels.
- 4) The authors need to clarify the correlation between FGF19 levels in blood, adjacent tissue and tumors studied.

Minor comments:

- 1) In the in vitro experiments with isolated hepatocytes the results regarding downstream pERK signaling activation are not conclusive. In figure 3d, pERK does not seem to increase with FGF19 concentrations added. Furthermore, data regarding pERK status is missing in figure 1e. Overall, the authors should demonstrate FGF19/FGFR4 signaling cascade activation in contrast to STAT3 activation in hepatocytes in the in vitro studies.
- 2) The authors conclude that IL-6 produced in the liver microenvironment promotes hepatocyte proliferation and HCC development. To further elucidate this mechanism, the authors should evaluate at least in vitro if the combination of IL6 and FGF19 induces hepatocyte proliferation, in contrast to FGF19 alone (results figure 4f and 4g).
- 3) For translational application and to further strengthen the potential interplay between hepatocytes and non-parenchymal microenvironment suggested in FGF19-induced in hepatocarcinogenesis, the authors should evaluate the correlation between FGF19 and IL6/STAT3 signaling in HCC patients.
- 4) In figure 1g the data regarding Tumor area in hepatocyte deficient STAT3 mice is missing.
- 5) Given the magnitude of differences of some of the data presented I would recommend the authors to introduce broken axis graphs for the clear representation of data in control mice in comparison to the model used.
- 6) In figure 6d, data regarding tumor area in IL6-/- mice with FGF19 expression is missing. There are 2 mice with tumors (figure 1b and 1c).
- 7) In supplementary figure 4a, the IHC images are not clear and PCNA and Ki67 staining cannot be visualized.

Reviewer #2 (Remarks to the Author):

Review for Zhou et al, "Activation of IL-6-Stat3 Signaling mediates FGF19-driven hepatocarcinogenesis".

This is a very solid paper with huge amounts of well organized and convincing data. It almost seems like that paper was previously prepared for a higher profile journal.

The message is new and exciting- you can preserve "metabolic improvement" functions of FGF19, while finding a mechanism (IL-6/Stat3). The data is really solid and involves a lot of in vivo studies with all of the proper controls. Authors also combine genetic approaches and pharmacological approaches to disrupt pathways of interest and prove the proposed mechanisms.

I really want to pick on something but this paper is almost perfect. Only few comments and suggestions:

1) Authors should be very upfront in the discussion that the fact that Stat3 inactivation and IL-6 neutralization results may also mean that IL-6 and Stat3 are extremely important in HCC beyond FGF19 signaling. In this case you would also see a decrease in HCC in mice which are not treated with FGF19. The only problem is that in this particular model these "naive/untreated" mice do not develop any tumors (see Fig 1 E, F, Fig 6 etc). To make it to absurd point if you knockout something important (Stat3, Il-6) like b-actin, for sure tumors will not develop, but that will not fully mean that it is because of FGF19-> b-actin pathway . I would suggest couple of sentences into Discussion section addressing this point. Otherwise these results look really convincing and strong.

2) Suppl. Fig 5 and other figures representing IL-6 producing cells look a little bit strange. All IL-6+ cells look overwhelmingly positive for many markers which sometimes do not do together, like Ly6G together with F4/80 and together with NK1.1. Could you please check all of the compensations- APC color seems to co-stain with other colors closer to APC channel, like A700, A780/APCCy7 and PECy7. It is convincing that T and B cells do not express IL-6 in response to restimulation with FGF19 though.

Reviewer #3 (Remarks to the Author):

The authors examined the role of STAT3 during the induction of liver cancer (HCC) in mice over-expressing FGF19. They provide evidence that FGF19 induces IL6 expression in non-parenchymal cells in the liver, which subsequently activates STAT3 in hepatocytes to induce downstream target gene expression. Their data from knockout mice indicate that both IL6 and STAT3 are required for the carcinogenic effects of FGF19 in liver but not for the metabolic effects. They go on to show that IL6 expression is likely confined to cells of hematopoietic origin, and they show that inhibition of IL6/STAT3 signaling (either with small molecules, downstream negative regulators, or antibodies) results in a therapeutic effect against HCC induction. Overall, the data represent a novel finding of IL6/STAT3 signaling in FGF19-induced cancer, similar to the role of this pathway in a large number of other tumors.

While the data are strongly suggestive of a cell non-autonomous pathway for IL6 production, this conclusion is based largely on in vitro experiments. It is known that cultured primary hepatocytes lose many of the characteristics reflected in an intact liver, so finding that FGF19 fails to activate STAT3 in vitro can only be considered supportive of a non-cell autonomous effect, not proof. A more definitive experiment would be to use chimeric mice in which wild type mice are reconstituted with an IL6-deficient hematopoietic system and then test whether STAT3 phosphorylation and tumorigenesis are lost in the wild type livers following AAC-FGF19 infection.

Similarly, the data shown in Fig. 5 would be more convincing if analysis of control non-parenchymal cells were included in the FACS measurements (e.g., cells from livers of mice that have not been exposed FGF19).

Point-by-point reply to Reviewers (Authors' response in blue):

Reviewers' comments are italicized

Reviewer #1: (Remarks to the Author):

In the present manuscript the authors aim to characterize the mechanism of FGF19-dependent tumor initiation through a non-cell-autonomous activation of IL6/STAT3 signaling pathway. The study adds information regarding the mechanism of FGF19-induced hepatocarcinogenesis, particularly in relation to JAK/STAT3 signaling activation. Nonetheless several issues remain to be clarified.

We thank the reviewer for evaluating and providing constructive feedback on our work. We provide all of the requested data and/or clarification in a point-by-point manner below to address the reviewer's concerns and believe that the revised manuscript should address the entirety of the reviewer's comments. The changes are highlighted in the revised manuscript text, and also include a number of additions to the original manuscript: a new figure (Figure 8), a new supplementary figure (Supplementary Figure 8), 6 new supplementary tables (Supplementary Tables 1-6), and multiple new panels to existing figures (Figure 5h, Supplementary Figure 1b-c, Supplementary Figure 5c-e, Supplementary Figure 6b).

Major comments

1) Human data: The authors state that FGF19 overexpression would lead to STAT3 activation in HCC. From a translational perspective, this hypothesis requires to be confirmed in human samples.

This is a highly relevant point and we agree with the reviewer's request that the translational relevance of our finding is an important issue to address. To that end, we conducted additional studies to strengthen the manuscript. Specifically, we present a new analysis of gene expression data from the Cancer Genome Atlas (TCGA) database, comprising more than 11000 samples across 33 different cancer entities, and show that FGF19 expression correlates with the expression of STAT3 target genes, including BIRC5, BCL2, HSPA4, and BCL2L1. These new data are included in Supplementary Figure 8d of the revised manuscript.

Furthermore, we performed *in situ* hybridization on 83 human HCC samples using RNAscope duplex chromogenic technology to simultaneously stain FGF19 and BIRC5, a STAT3 target gene. We identified 10 out of 83 HCC specimens overexpressing FGF19. Importantly, concomitant FGF19 and BIRC overexpression was evident in all 10 of these samples, establishing a positive correlation between the expression of FGF19 and a STAT3 target gene in human FFPE HCC specimens. These new data have been incorporated in the revised version of the manuscript (Figure 8e).

We amended the Results section to describe these new results (pages 17-19), and updated Methods and Figure Legends accordingly. All available clinical data of patients, including gender, age, diagnosis, tumor stage, etc. are summarized in Supplementary Tables 1-6. We hope these additional data address the questions regarding the potential translation of our findings, and that they serve to strengthen the manuscript.

2) Mechanism of induction of IL6 overexpression. The authors state that FGF19

overexpression induces release of IL6 from the innate immune cells from the microenvironment. The authors need to clarify if this is occurring in the microenvironment of the tumor or just of the adjacent tissue. This reviewer is concerned by the fact that the information about IL6 overexpression in the tumor is lacking.

In our study, we found that FGF19 induces IL-6 released from innate immune cells in the liver microenvironment, not tumor microenvironment. This effect occurs in an acute setting (two hours following FGF19 dosing), prior to tumor development.

With respect to the detailed mechanisms underlying FGF19-induced IL-6 expression, a complete investigation of the molecular mechanism is beyond the scope of the present study. Identifying innate immune cell subsets which express FGFR4 and KLB receptors, and interrogating the functions of these receptors by cell type-specific knockout, will be addressed in future studies.

3)Is tumor proliferation related to STAT 3 activation? The study is focused on activation of STAT3 signaling as the main driver of FGF19 –induced tumorigenesis. However, the model describes activation of WNT/beta-catenin signaling, and translocation of this protein into the nucleus. Can the authors explore if this might be also a mechanism of proliferation, and whether there is a direct correlation with FGF19 levels.

We have added new immunohistochemical data (Supplementary Figure 1c) showing that FGF19 increased the levels of the proliferative marker, Ki-67, whose expression was eliminated by hepatocyte-specific deletion of STAT3. These data are consistent with our model that FGF19-induced tumor proliferation requires STAT3 activation.

Regarding WNT/ β -catenin activation and translocation, Pai and colleagues have shown that FGF19 increases tyrosine phosphorylation of β -catenin in cancer cell lines (*Cancer Res* 2008, 68:5086). Additionally, Latasa and colleagues have suggested that FGF19 could also activate β -catenin via crosstalk with EGFR pathway (*PLoS ONE* 2012, 7:e52711). Our study does not exclude other mechanisms by which FGF19 may promote tumorigenesis, and the non-cell-autonomous signaling described in this report could work in parallel with such pathways. More importantly, our results provide an additional strategy for therapeutic intervention. We thank the reviewer for bringing to our attention these additional mechanisms, and have expanded the discussion to include reference to these relevant studies (page 23, lines 7-9).

4)The authors need to clarify the correlation between FGF19 levels in blood, adjacent tissue and tumors studied.

In order to address the point raised by the reviewer, we have performed additional experiments to measure FGF19 levels in blood, tumors and adjacent tissues. As shown in Supplementary Figure 8c, there is a clear correlation between blood and tumor levels of FGF19 in mouse models. However, there is no significant difference between FGF19 levels in tumors and adjacent liver tissues. Interestingly, similar results were observed in human HCC samples in TCGA database (box plots presented in Figure 8c, showing elevated FGF19 expression in tumor and adjacent non-tumor tissues versus normal liver samples, but no significant differences in

FGF19 levels between tumors and adjacent non-tumor tissues).

Minor comments

1) In the in vitro experiments with isolated hepatocytes the results regarding downstream pERK signaling activation are not conclusive. In figure 3d, pERK does not seem to increase with FGF19 concentrations added. Furthermore, data regarding pERK status is missing in figure 3e. Overall, the authors should demonstrate FGF19/FGFR4 signaling cascade activation in contrast to STAT3 activation in hepatocytes in the in vitro studies.

pERK signaling is clearly activated in response to FGF19 treatment in cultured mouse hepatocytes *in vitro* in Figure 3d. We feel that this comment about pERK might derive from a misunderstanding of Figure 3d, which is a time course study, not dose escalation.

We apologize for missing pERK immunoblot in the original Figure 3e, and have included these data in the revised Figure 3e. Our results showed that pERK signals are rapidly activated 5 minutes following FGF19 addition in isolated hepatocytes of murine or human origins, which contrasts with the lack of pSTAT3 activation by FGF19 in same cells.

2) The authors conclude that IL-6 produced in the liver microenvironment promotes hepatocyte proliferation and HCC development. To further elucidate this mechanism, the authors should evaluate at least in vitro if the combination of IL6 and FGF19 induces hepatocyte proliferation, in contrast to FGF19 alone (results figure 4f and 4g).

IL-6 alone did not increase BrdU incorporation in isolated hepatocytes, nor did it exert additive or synergistic effects with FGF19.

3) For translational application and to further strengthen the potential interplay between hepatocytes and non-parenchymal microenvironment suggested in FGF19-induced in hepatocarcinogenesis, the authors should evaluate the correlation between FGF19 and IL6/STAT3 signaling in HCC patients.

Please see our response to the reviewer's Major Comment #1 regarding the need for human data.

4) In figure 1g the data regarding Tumor area in hepatocyte deficient STAT3 mice is missing.

The reviewer's comment about tumor area might derive from a misunderstanding of Figure 1g. Multiple previous reports have shown that FGF19-induced tumors are invariably stained positive for glutamine synthetase (*Nicholes et al., Am J Pathol 2002, 160:2295; Zhou et al., Cancer Res 2014, 74:3306; Luo et al., Sci Transl Med 2014, 6:247ra100*). As stated in Supplementary Methods, only glutamine synthetase-positive tumor areas were quantified for morphometric analysis using ImagePro software. The tumor area in Figure 1g for FGF19-STAT3^{ΔHep} mice is zero, not missing. The only tumor nodule detected in FGF19-STAT3^{ΔHep} mice (Figures 1e and 1f) was glutamine synthetase-negative. Nonetheless, to address the reviewer's concern, we have added a new panel to Supplementary Figure 1 (panel 1b) showing individual mouse liver images stained with anti-glutamine synthetase in

support of this conclusion.

5) Given the magnitude of differences of some of the data presented I would recommend the authors to introduce broken axis graphs for the clear representation of data in control mice in comparison to the model used.

Please see our response to above Minor Comment #4. Since some groups have tumor area of zero in multiple figures/panels (Figures 1g, 6d, 7b, 7e, 7i), or FGF19 concentrations of zero (Figures 1i, 4e, 6f), we feel that broken axis might not be necessary for these graphs.

6) In figure 6d, data regarding tumor area in *IL6*^{-/-} mice with FGF19 expression is missing. There are 2 mice with tumors (figure 6b and 6c).

Please see our response to above Minor Comment #4. The tumor area in Figure 6d for FGF19-*IL6*^{-/-} mice is zero, not missing. The two tumor nodules in FGF19-*IL6*^{-/-} mice (Figures 6b and 6c) were determined to be glutamine synthetase-negative. In response to the reviewer's comment, we have added new panels to Supplementary Figures 6 (panel 6b) showing images of individual mouse liver sections stained with anti-glutamine synthetase.

7) In supplementary figure 4a, the IHC images are not clear and PCNA and Ki67 staining cannot be visualized.

We thank the reviewer for making us aware of this issue, and have replaced Supplementary Figure 4a with images of higher resolutions.

Reviewer #2 (Remarks to the Author):

Review for Zhou et al, "Activation of IL-6-Stat3 Signaling mediates FGF19-driven hepatocarcinogenesis".

This is a very solid paper with huge amounts of well-organized and convincing data. It almost seems like that paper was previously prepared for a higher profile journal. The message is new and exciting- you can preserve "metabolic improvement" functions of FGF19, while finding a mechanism (IL-6/Stat3). The data is really solid and involves a lot of in vivo studies with all of the proper controls. Authors also combine genetic approaches and pharmacological approaches to disrupt pathways of interest and prove the proposed mechanisms.

We appreciate that the reviewer recognizes the importance and novelty of our work, and thank the reviewer for the supportive feedback.

I really want to pick on something but this paper is almost perfect. Only few comments and suggestions:

1) Authors should be very upfront in the discussion that the fact that Stat3 inactivation and IL-6 neutralization results may also mean that IL-6 and Stat3 are extremely important in HCC beyond FGF19 signaling. In this case you would also see a decrease in HCC in mice which are not treated with FGF19. The only problem is that in this particular model these "naïve/untreated" mice do not develop any tumors (see Fig 1 E, F, Fig 6 etc). To make it to absurd point if you knockout something important (Stat3, Il-6) like b-actin, for sure tumors will not develop, but that will not fully mean

that it is because of FGF19-> b-actin pathway. I would suggest couple of sentences into Discussion section addressing this point. Otherwise these results look really convincing and strong.

We are aware that the IL-6/STAT3 signaling is important for HCC development and agree with the reviewer that we should explicitly acknowledge this point in the discussion. In particular, multiple recent reports suggest the importance of IL-6/STAT3 in liver cancer stemness pathway, which could serve as a nodal point in hepatocarcinogenesis in general (Wan *et al. Gastroenterology* 2014, 147:1393; He *et al., Cell* 2013, 155:384). We have now expanded and reorganized the Discussion so that text describing the essential roles of STAT3/IL-6 in HCC development now appears near the beginning of the section (pages 20-21).

2) Suppl. Fig 5 and other figures representing IL-6 producing cells look a little bit strange. All IL-6+ cells look overwhelmingly positive for many markers which sometimes do not do together, like Ly6G together with F4/80 and together with NK1.1. Could you please check all of the compensations- APC color seems to co-stain with other colors closer to APC channel, like A700, A780/APCCy7 and PECy7. It is convincing that T and B cells do not express IL-6 in response to restimulation with FGF19 though.

The reviewer raised a valid concern about multicolor flow cytometry analysis. Studies described in Figures 5f, 5g and Supplementary Figures 5b,5c were conducted after optimizing fluorescence compensation settings with CompBead (BD Biosciences). Specifically, CompBead particles were mixed with fluorochrome-conjugated antibodies to provide distinct positive and negative stained populations, and were then used to set compensation levels with the automated compensation setup available in FACSDiva™ software. This method allowed us to more accurately establish compensation corrections for spectral overlap for combination of multiple fluorochrome-labeled antibodies, especially tandem dye conjugates such as PE-Cy7 and APC-Cy7. However, CompBead particles may not provide correct compensation due to spectral overlap between the V500- and AmCyan-conjugated antibodies, which we chose to avoid in our studies. In hopes of clarifying the data, we have now added this information in the updated Supplementary Methods (Supplementary Information, page 50, lines 13-16).

Furthermore, we conducted additional experiments in which we selectively depleted immune cell subsets, specifically and individually eliminating Kupffer cells (via clodronate liposomes by intravenous injection), neutrophils (via anti-Ly-6G antibodies), and CD8+ T cells (via anti-CD8 antibodies). We provide new data presented in Figure 5h and Supplementary Figures 5d-e showing that depletion of Kupffer cells reduced pSTAT3^{Y705} signals by FGF19 treatment, while depletion of neutrophils or CD8+ T cells had no obvious impact. These results suggest that Kupffer cell-produced IL-6 contributes, at least in part, to FGF19-induced STAT3 phosphorylation.

Reviewer #3 (Remarks to the Author):

The authors examined the role of STAT3 during the induction of liver cancer (HCC) in mice over-expressing FGF19. They provide evidence that FGF19 induces IL6 expression in non-parychymal cells in the liver, which subsequently activates STAT3 in hepatocytes to induce downstream target gene expression. Their data from knockout mice indicate that both IL6 and STAT3 are required for the carcinogenic effects of FGF19 in liver but not for the metabolic effects. They go on to show that IL6 expression is likely confined to cells of hematopoietic origin, and they show that inhibition of IL6/STAT3 signaling (either with small molecules, downstream negative regulators, or antibodies) results in a therapeutic effect against HCC induction. Overall, the data represent a novel finding of IL6/STAT3 signaling in FGF19-induced cancer, similar to the role of this pathway in a large number of other tumors.

While the data are strongly suggestive of a cell non-autonomous pathway for IL6 production, this conclusion is based largely on in vitro experiments. It is known that cultured primary hepatocytes lose many of the characteristics reflected in an intact liver, so finding that FGF19 fails to activate STAT3 in vitro can only be considered supportive of a non-cell autonomous effect, not proof. A more definitive experiment would be to use chimeric mice in which wild type mice are reconstituted with an IL6-deficient hematopoietic system and then test whether STAT3 phosphorylation and tumorigenesis are lost in the wild type livers following AAV-FGF19 infection.

We fully agree with the reviewer that the *in vitro* experiments do not exclude the possibility that a cell-autonomous mechanism may still be relevant, especially in tumor cells during malignant transformation. We have expanded the discussion to describe the possibility of cell-autonomous production of IL-6 by hepatocytes in tumors (page 23, lines 20-23).

The reviewer made a sound suggestion regarding the use of chimeric mice bone marrow transplant experiments to further delineate mechanisms. However, several recent studies have shown that bone marrow transplant of hematopoietic stem cells may not fully replenish resident immune cells, such as liver-resident macrophage Kupffer cells (Schulz *et al.*, *Science* 2012, 336:86; Hashimoto *et al.*, *Immunity* 2013, 38:792; Ginhoux *et al.*, *Science* 2010, 330:841). Therefore, we elected to use an alternative strategy involving selective depletion of subpopulations of immune cells, specifically and individually eliminating Kupffer cells (via clodronate liposomes by intravenous injection), neutrophils (via anti-Ly-6G antibodies), and CD8+ T cells (via anti-CD8 antibodies). We present this new data in Figure 5h and Supplementary Figures 5d-e, showing that depletion of Kupffer cells, but not neutrophils or CD8+ T cells, reduced pSTAT3^{Y705} signals by FGF19 treatment. These results suggest that Kupffer cell-produced IL-6 contributes, at least in part, to FGF19-induced STAT3 phosphorylation.

Unfortunately, we did not conduct long-term tumorigenicity study in these depletion experiments. Due to time constraints (the time frame of HCC development in these mice is 6-12 months following AAV-FGF19 injection), we would not be able to comply with the 3-month revision term if we were to initiate a new study with AAV-FGF19. In evaluating the evidence we present in the revised manuscript, we hope that the reviewer would consider the totality of our data in multiple mouse models. A complete investigation of the molecular mechanism is beyond the scope of the present study.

Similarly, the data shown in Fig. 5 would be more convincing if analysis of control non-parenchymal cells were included in the FACS measurements (e.g., cell from livers of mice that have not been exposed FGF19).

As suggested by the reviewer, we have included data from non-parenchymal cells treated with vehicle (not exposed to FGF19), and updated Supplementary Figure 5c accordingly.

Reviewers' comments:

Reviewer #1 (Remarks to the Author):

Overall, the authors have successfully addressed the reviewers' concerns. However, there are still minor points that need clarification:

-The authors have incorporated the measurement of FGF19 levels in blood, tumors and adjacent tissue. They need to provide the correlation status between these levels, is there significant association between liver and blood levels?.

-Results regarding the tumor area in IL6-/- mice without FGF19 expression (figure 6d) have been clarified by the authors' explanation. However, they should specify that they only measure glutamine synthetase-positive tumors.

-The immunohistochemical staining of Ki67 (supplementary figure 1c) is a good method for confirming the role of STAT3 in tumor proliferation. However, the authors should provide a quantification of the staining.

-The answer to the Reviewer #1 Minor Comment 2 is confusing as it contradicts the hypothesis supported in this manuscript. Therefore, the authors should provide further information to explain this phenomenon.

-The image resolution of supplementary figure 4a is still not sufficient.

-Supplementary figures 6c-e are missing.

Reviewer #2 (Remarks to the Author):

Authors have sufficiently addressed prior critique.

Reviewer #3 (Remarks to the Author):

The authors have made substantial improvements to this manuscript by inclusion of additional data. In particular, they now provided additional evidence for a non-cell autonomous mechanism for IL6 production following FGF treatment. However, there is one new dataset that is less than convincing, relating to the potential implications of these experimental animal findings to human cancer. In new data, the authors provide TCGA analysis to suggest that FGF19 expression correlates with a STAT3 gene expression signature. However, the level of analysis provided is insufficient to support the authors' conclusions. The target genes used in the study are not definitive markers of IL6/STAT3 signaling, since they are regulated by numerous cancer-related processes (wnt, p53, etc), and may simply reinforce the conclusion that FGF19 correlates with malignancy rather than providing strong evidence for involvement of STAT3. The conclusion that the authors wish to reach would require documenting that a broad STAT3 signature correlates with FGF19. A more compelling data set to augment Fig. 8e would include analysis of STAT3 targets in FGF19-free tumors and of activated STAT3 (phospho-tyrosine or elevated expression).

Point-by-point reply to Reviewers (Authors' response in green):

Reviewers' comments are italicized

Reviewer #1 (Remarks to the Author):

Overall, the authors have successfully addressed the reviewers' concerns. However, there are still minor points that need clarification:

-The authors have incorporated the measurement of FGF19 levels in blood, tumors and adjacent tissue. They need to provide the correlation status between these levels, is there significant association between liver and blood levels?

Yes, there is significant association between liver and blood FGF19 levels in mice injected with AAV-FGF19. These data are presented in Supplementary Figure 8c (right panel) with correlation coefficient value added on chart.

-Results regarding the tumor area in IL6-/- mice without FGF19 expression (figure 6d) have been clarified by the authors' explanation. However, they should specify that they only measure glutamine synthetase-positive tumors.

Per the reviewer's suggestion, we have now explicitly specified glutamine synthetase-positive tumors on page 6 (line 6), page 14 (lines 3 and 5), and in Supplementary Methods (page 48, lines 19-20 of Supplementary Information).

-The immunohistochemical staining of Ki67 (supplementary figure 1c) is a good method for confirming the role of STAT3 in tumor proliferation. However, the authors should provide a quantification of the staining.

We have added quantification of Ki-67-positive cells in the revised Supplementary Figure 1c as a bar graph, and updated Supplementary Methods on definition of Ki-67 labeling index (page 48, lines 20-22 of Supplementary Information).

-The answer to the Reviewer #1 Minor Comment 2 is confusing as it contradicts the hypothesis supported in this manuscript. Therefore, the authors should provide further information to explain this phenomenon.

IL-6 signaling has been identified as a key pathway promoting the expansion of liver cancer progenitors and/or cancer stem cells (*He et al., Cell 2013, 155:384; Wan et al., Gastroenterology 2014, 147:1393*). These liver tumor-initiating cells represent a rare population of liver cells and express surface markers such as CD24 (*Lee et al., Cell Stem Cell 2011, 9:50*), CD90 (*Yang et al., Cancer Cell 2008, 13:153*), CD133 (*Ma et al., Cell Stem Cell 2010, 7:694*) or EpCAM (*Yamashita et al., Gastroenterology 2009, 136:1012*). We have amended the Discussion section (page 21, lines 10-14) to include these information as a possible explanation for the lack of proliferative effects of IL-6 (or in combination with FGF19) on cultured hepatocytes, which are not enriched for specific stem cell subpopulations.

-The image resolution of supplementary figure 4a is still not sufficient.

We have replaced images in Supplementary Figure 4a with images of higher resolution.

-Supplementary figures 6c-e are missing.

We are very grateful that the reviewer brought this oversight to our attention, and have made the corrections.

Reviewer #2 (Remarks to the Author):

Authors have sufficiently addressed prior critique.

We thank the reviewer for his/her thoughtful appraisal and constructive suggestions.

Reviewer #3 (Remarks to the Author):

The authors have made substantial improvements to this manuscript by inclusion of additional data. In particular, they now provided additional evidence for a non-cell autonomous mechanism for IL6 production following FGF treatment. However, there is one new dataset that is less than convincing, relating to the potential implications of these experimental animal findings to human cancer. In new data, the authors provide TCGA analysis to suggest that FGF19 expression correlates with a STAT3 gene expression signature. However, the level of analysis provided is insufficient to support the authors' conclusions. The target genes used in the study are not definitive markers of IL6/STAT3 signaling, since they are regulated by numerous cancer-related processes (wnt, p53, etc), and may simply reinforce the conclusion that FGF19 correlates with malignancy rather than providing strong evidence for involvement of STAT3. The conclusion that the authors wish to reach would require documenting that a broad STAT3 signature correlates with FGF19. A more compelling data set to augment Fig. 8e would include analysis of STAT3 targets in FGF19-free tumors and of activated STAT3 (phospho-tyrosine or elevated expression).

We agree with the reviewer that some markers (such as *CCND1*) we used may not be definitive for IL-6/STAT3 signaling, and are regulated by other cancer-related pathways. It is a valid concern raised by the reviewer. We have tried staining human HCC samples with antibodies against phosphorylated STAT3, but low sensitivity of the assay precluded drawing a definitive conclusion.

As an alternative approach, we analyzed a broader set of STAT3 signature genes (*BIRC5*, *BCL2*, *HSPA4*, *MCL1*, *BCL2L1*) on FGF19-expressing HCCs and confirmed the upregulation of these genes by quantitative RT-PCR. We have provided these new data as Figure 8f and Supplementary Table 7, and amended Results (page 19, lines 7-9), Methods (page 26, lines 19-21), and Figure Legends (page 43, lines 11-14) accordingly.

Finally, we appreciate the reviewer's comments on translatability, and acknowledge that the extent to which our observations in mice translate to human remains unclear, and have updated Discussion to remind readers to use caution when extrapolating results from animal models to human (page 24, lines 16-17).

REVIEWERS' COMMENTS:

Reviewer #1 (Remarks to the Author):

The authors have satisfactorily addressed all the reviewers' concerns and the manuscript has been improved by the addition of the requested data.

Reviewer #3 (Remarks to the Author):

The authors have made an excellent attempt to address the previous critique. The only previous point that remains insufficient is the analysis of human tumors for a correlation between FGF19 expression and an IL6/STAT3 signature. The authors provide data that demonstrate that a group of potential STAT3 target genes are expressed at higher levels in tumors also expressing FGF19, compared to normal tissues. This analysis does not address whether this gene expression profile is due to FGF19 or due to the malignant state of the samples. A more definitive conclusion could be reached by examining the expression of this gene signature in HCC samples lacking FGF19 expression.

Point-by-point reply to Reviewers (Authors' response in green):

Reviewers' comments are italicized

Reviewer #3 (Remarks to the Author):

The authors have made an excellent attempt to address the previous critique. The only previous point that remains insufficient is the analysis of human tumors for a correlation between FGF19 expression and an IL6/STAT3 signature. The authors provide data that demonstrate that a group of potential STAT3 target genes are expressed at higher levels in tumors also expressing FGF19, compared to normal tissues. This analysis does not address whether this gene expression profile is due to FGF19 or due to the malignant state of the samples. A more definitive conclusion could be reached by examining the expression of this gene signature in HCC samples lacking FGF19 expression.

Per reviewer's suggestion, we have now provided additional data in the revised Figure 8f to include human HCC samples lacking FGF19 expression, and updated Results (page 19, lines 9-10), Methods (page 26, lines 20-21), and Supplementary Table 3 (please note that supplementary tables were re-ordered and this table was previously named Supplementary Table 7) accordingly. In summary, our data support the expression of STAT3 signature (BIRC5, BCL2, HSPA4) in HCC samples over-expressing FGF19, but not in HCC samples lacking FGF19 expression.

We hope that these new data address the remaining concern of the reviewer, and would like to thank the reviewer for his/her suggestions to help us improve this manuscript.